# Common synaptic phenotypes arising from diverse mutations in the human NMDA receptor subunit GluN2A

Marwa Elmasri [1], Daniel William Hunter[1], Giles Winchester [1], Ella Emine Bates[1], Wajeeha Aziz[1], Does Moolenaar Van Der Does[1], Eirini Karachaliou[1], Kenji Sakimura[2] & Andrew. Charles Penn [1✉]

Dominant mutations in the human gene *GRIN2A*, encoding NMDA receptor (NMDAR) subunit GluN2A, make a significant and growing contribution to the catalogue of published single-gene epilepsies. Understanding the disease mechanism in these epilepsy patients is complicated by the surprising diversity of effects that the mutations have on NMDARs. Here we have examined the cell-autonomous effect of five GluN2A mutations, 3 loss-of-function and 2 gain-of-function, on evoked NMDAR-mediated synaptic currents (NMDA-EPSCs) in CA1 pyramidal neurons in cultured hippocampal slices. Despite the mutants differing in their functional incorporation at synapses, prolonged NMDA-EPSC current decays (with only marginal changes in charge transfer) were a common effect for both gain- and loss-of-function mutants. Modelling NMDA-EPSCs with mutant properties in a CA1 neuron revealed that the effect of *GRIN2A* mutations can lead to abnormal temporal integration and spine calcium dynamics during trains of concerted synaptic activity. Investigations beyond establishing the molecular defects of GluN2A mutants are much needed to understand their impact on synaptic transmission.

[1] Sussex Neuroscience, School of Life Sciences, University of Sussex, Brighton BN1 9QG, UK. [2] Department of Cellular Neurobiology, Brain Research Institute, Niigata University, Niigata 951-8585, Japan. ✉email: A.C.Penn@sussex.ac.uk

N-methyl-D-aspartate receptors (NMDARs) are conserved synaptic proteins underling the late component of post-synaptic potentials at excitatory synapses[1]. The receptor plays key roles in sculpting nervous system development[2–4] and spine morphology[5], shaping temporal integration of synaptic potentials[6–8], regulating synaptic efficacy[9–11], and the formation of new declarative memories[12]. Core to NMDARs achieving these roles is their high-calcium permeability, depolarisation-dependent unblock of extracellular magnesium ions from the channel pore, and the slow time course of NMDAR channel closure (deactivation) following brief exposure to neurotransmitter glutamate[1,10,13]. These properties are determined to a large extent by subunit composition and stoichiometry of NMDARs[14]. In adulthood, conventional NMDARs are most frequently heterotetramers composed of obligatory GluN1 subunits and regulatory GluN2 subunits, while unconventional glycine-gated GluN1/3 heterotetramers are transiently expressed during early postnatal development[10,15], persisting into adulthood only in the Medial Habenula[16]. The GluN2 subunits contain the ligand-binding site indispensable for the glutamate-gated operation of NMDARs at excitatory synapses[17]. Remarkably, four GluN2 subunits, suffixed by A–D, account for much of the diversity observed for a wide range of functional characteristics of NMDARs[10,18,19].

NMDAR subunit expression varies between neuronal types and during development thus providing of means of adapting the NMDAR-mediated component of excitatory postsynaptic potentials (EPSPs) to the physiological context. Principal neurons account for the majority of neurons in the adult forebrain, where they predominantly express GluN2A and 2B, and to a much lesser extent GluN2C or D[20–22]. While GluN2B predominates at birth and continues to make a major contribution to synaptic NMDARs into adulthood, GluN2A is upregulated during early postnatal development[20,23–26], by neuronal activity[27–29] and sensory experience[30–33], and accounts for much of the observed developmental speeding of NMDAR-mediated excitatory postsynaptic currents (NMDA-EPSCs)[23,24]. Mice with germline genetic knockout of the GluN2A gene (called grin2a) are incapable of undergoing the subunit switch and have impaired spatial learning[34], altered vocalisations and spontaneous epileptiform discharges[35,36]. Their learning defects are also characterised by raised thresholds for the induction of contextual fear learning and long-term synaptic potentiation[37].

The significance of NMDARs is well-demonstrated by the range of clinical phenotypes of patients with mutations in the genes encoding NMDAR subunits. Since the development of next-generation sequencing methods in the mid-1990s there has been a steady rise in the number of epilepsy-associated genes[38] with the addition of the human GluN2A receptor gene, GRIN2A, to this list only a decade ago[39–41]. Published figures recently showed that >250 patients are known to have GRIN2A mutations, either de novo or transmitted with dominant inheritance[42,43]. Epilepsy occurs in about 90% of these patients[43,44], with GRIN2A mutations representing the best known genetic cause of epilepsy-aphasia spectrum (EAS) disorders, where they account for 9–20% of cases[42,45–49]. The clinical phenotype of EAS varies considerably from the more common and benign childhood (Rolandic) epilepsy with centrotemporal spikes (CECTS), to the rarer and more severe Landau–Kleffner syndrome (LKS) and continuous spike-and-wave during slow-wave sleep syndrome (CSWSS)[42,50]. Although rare, the severe epilepsy cases are often also accompanied by progressive cerebral dysfunction (epileptic encephalopathy) and are frequently comorbid with intellectual disability and language disorders[43]. The response to existing treatments in patients with severe EAS can be unpredictable and no anti-epileptic drug (AED) has been demonstrated to be effective specifically in GRIN2A-related disorder[42].

Most research on GRIN2A mutations has investigated the effects of missense mutations on GluN2A protein function in heterologous expression systems[45,47,51–58]. A common outcome of these studies is that epilepsy-associated GRIN2A mutations have strikingly variable functional consequences on NMDARs[51,52,57,59]. Until recently, the impact of GluN2A mutations on the NMDAR-mediated component of synaptic transmission has received relatively little attention[60]. While modelling of synaptic charge transfer has been made for some GRIN2A mutations[57], these models do not consider how expressing the mutant GluN2A receptor could feedback on synaptic transmission[7], or to what extent and how other native NMDAR subunits continue to support NMDAR-mediated synaptic transmission[23]. In this study, we have evaluated and compared the phenotypes and mechanisms of synaptic dysfunction in CA1 neurons expressing epilepsy-associated gain- and loss-of-function GRIN2A mutants.

## Results

**Contrasting gain- and loss-of-function epilepsy-associated missense mutations in GRIN2A.** More than 40% of mutations discovered in GRIN2A are missense mutations (Fig. 1a), with the majority of them (approx. 60%) being in the sequence encoding the ligand-binding domain (LBD) or the ion channel[43]. Missense mutations cause substitutions of amino acids and can vary considerably in their consequence on protein function and expression[45,47,51–58]. Considerable experimental data have been published characterising how disease-linked mutations in GRIN2A affect the surface expression and biophysical properties of GluN2A-containing NMDA receptors in heterologous expression systems, namely in HEK cells and Xenopus oocytes. We compiled data for 20 different mutations from various publications and databases[45,47,51–58] (Fig. 1b: blue spheres). The most comprehensive sets of measurements have been made for mutations in the LBD and it's linkers to the ion channel, which are the fundamental operating units for glutamate-gated opening of the channel[61,62] (Fig. 1b). The measured variables graphed in Fig. 1c reflect receptor activation (agonist potencies), channel activity (open probability and current deactivation), surface expression, or a combination of the above (current density). The effects of the mutations were up to 4 orders magnitude different from WT receptors with the effects being generally larger for glutamate potency and current density (Fig. 1c). Loss-of-function effects were most frequently observed for current density (79%), surface expression (77%) and open probability (63%), while receptor deactivation time was the only receptor property to be dominated by gain-of-function effects (59%) (Fig. 1c).

Although mutation effects can be easily identified for specific receptor properties (Fig. 1c), overall classification of mutations can become difficult as we illustrate with some examples. The R518H mutation was identified in a patient with CSWSS and was initially described as a gain-of-function mutation based on the longer open time of the channel in single-channel recordings[47]. Since then, others have characterised the mutation further and reported that the mutant is poorly expressed on the cell surface and gives negligible whole-cell currents[56,57], which raises uncertainty on the overall effect of this mutation. Another complicated example is the case of K669N. This mutant expresses normally and exhibits a 3–4-fold higher agonist potency and deactivation time constant compared to WT receptors[57]. However, the open probability of the channel is estimated to be 4–5 times lower making the overall impact of this mutation also unclear.

The examples described above motivated us to adopt factor analysis to assist in interpreting the dataset of GluN2A mutation

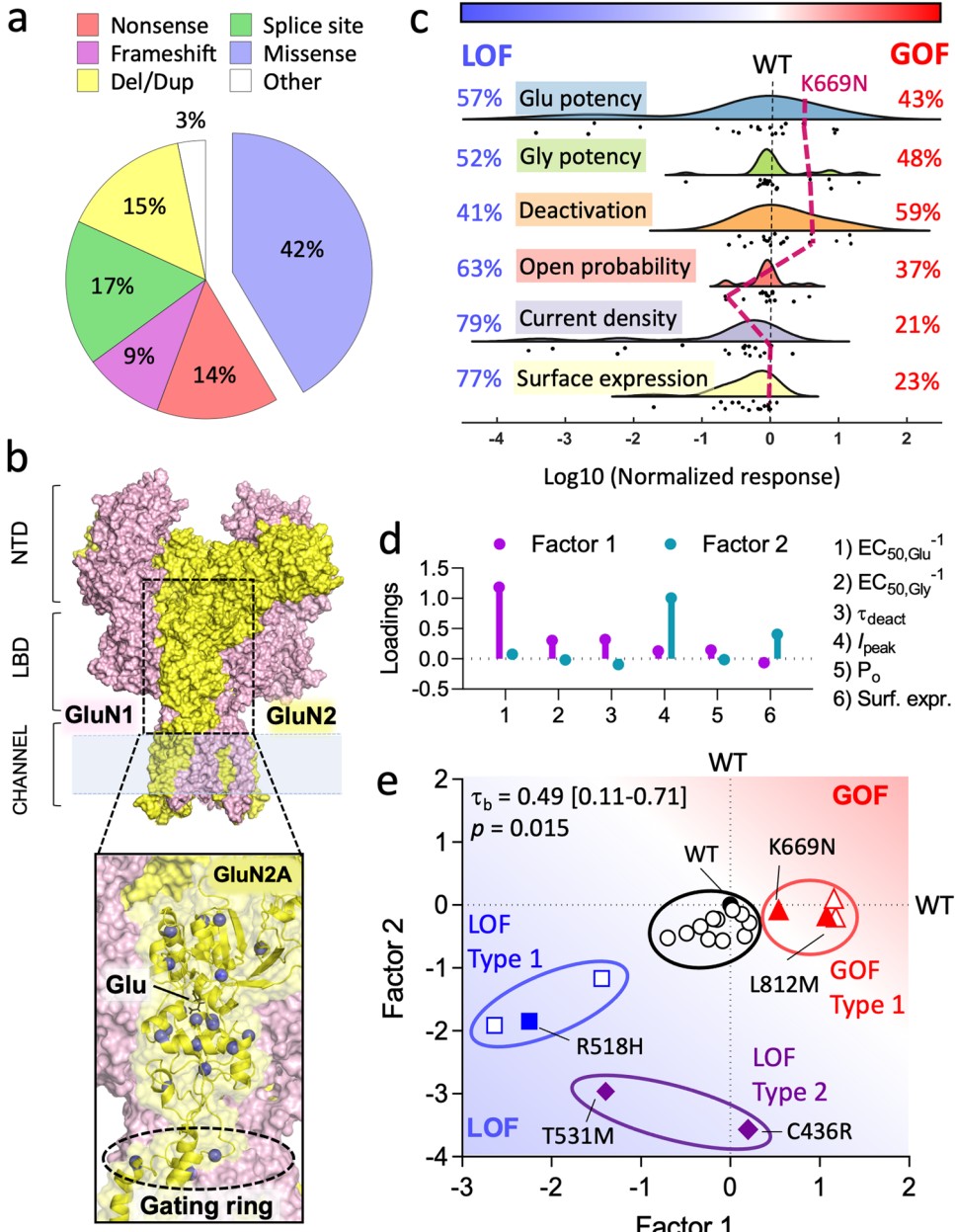

**Fig. 1 Missense *GRIN2A* mutations can impart severe overall gain or loss of function effects on NMDARs. a** A pie chart illustrating the proportion of different types of mutations in human *GRIN2A*. Missense mutations make over 40% of all *GRIN2A* mutations but their impact on NMDAR function can be hard to predict. **b** Surface representation of an NMDAR heterotetramer of GluN1 and GluN2 (pink and yellow respectively). The ligand-binding domain (LBD) of GluN2A is boxed out and magnified to illustrate the glutamate (Glu) binding site, the adjacent gating ring and the location of some functionally characterised disease-associated missense mutations (blue spheres)[45,47,51–58]. NTD = Amino terminal domain. **c** Raincloud plot showing quantification of the gain (>0) and loss (<0) of function of the mutations shown in (**b**) with respect to different properties measured for mutant GluN1/2A receptors expressed in HEK cells or oocytes[45,47,51–58]. The black dashed line at 0 corresponds to WT GluN2A. The tick dashed maroon line corresponds to the K669N mutation, which serves to illustrate that some mutations can be simultaneously gain-of-function (GOF), loss-of-function (LOF), or have no effect across different properties of the receptor. **d** Stem-and-leaf plot of the loadings (pattern matrix) of the first two components of a principal component analysis after oblique rotation. The loading for first factor is highest for glutamate potency ($EC_{50, Glu}^{-1}$), while the loading for the second factor is highest for peak current density ($I_{peak}$). **e** Scatter plot of the scores of the 20 *GRIN2A* mutations for each of the factors 1 and 2 from **d**. The graph is annotated with the results of K-means clustering: mutants that are GOF for factor 1 (red, Type 1 GOF); mutants that are mostly LOF for factor 1 (blue, Type 1 LOF); mutants that are LOF mostly for factor 2 (purple, Type 2 LOF); and WT-like mutants (black). Filled symbols represent mutations used for experiments in this study. The graph is annotated with Kendall's tau-b correlation coefficient accompanied by bootstrap confidence intervals and *p*-values (see methods). The blue-white-red gradient fill in figure panels (**c**) and (**e**) are purely qualitative to illustrate directionality of mutation effect (as LOF-WT-GOF, respectively).

effects (Fig. 1d, e). Principle component analysis provides a means to reduce multivariate data down to fewer, more manageable variables (components). The first two components calculated from the 6 variables collectively accounted for the majority of the variance (71 + 16 = 87%). After extraction and oblique rotation of these components, the first and second components (i.e., factors) had the highest loadings for glutamate potency (1/$EC_{50}$) and current density respectively (Fig. 1d).

Consistent with previous observations that GluN2 mutants with low glutamate potency also tend to exhibit poor surface expression[57], we found a moderate, but significant, positive correlation between the factors 1 and 2 (Fig. 1e).

The fragmented bivariate distribution of factor scores for the mutations suggests possible segregation of the mutants into different classes (Fig. 1e). Mutations with similar effects were classified by k-means clustering. The four clusters identified were: (a) gain-of-function mutants for factor 1 (4/20), referred to as GOF type 1; (b) loss-of-function mutants mostly for the factor 1 (3/20), referred to as LOF type 1; (c) severe loss-of-function mutants mostly for factor 2 (3/20), referred to as LOF type 2; and (d) WT-like mutations with overall little effect in one or both factors (11/20); (Fig. 1e). While the clustering of mutations is rather arbitrary and the wider population of mutations may exist rather as a continuum, the analysis serves to summarise and broadly classify this set of mutations. Given some of the assumptions and ambiguity in estimating glutamate potency for R518H and T531M (see methods), their cluster membership may not be accurate but it is clear that these represent mutations with a severe LOF phenotype. We cloned representative mutations from the different clusters that had most extensive functional characterisation that was published at the time (labelled in Fig. 1e), and we went on to examine their functional incorporation at synapses.

**Functional incorporation of gain-of-function, but not loss-of-function GluN2A mutants at synapses in CA1 neurons.** To assess functional incorporation of mutant GluN2A at excitatory synapses we have adopted a molecular replacement strategy in mouse organotypic hippocampal slices[63]. CA1 neurons in slices from adult mice are reported to express GluN2A and GluN2B subunits and their combined deletion effectively depletes excitatory synapses of NMDA receptors[23]. Conditional deletion of the native mouse grin2a and b alleles was achieved using grin2a$^{fl/fl}$ grin2b$^{fl/fl}$ mouse lines. In these mice, loxP sites are orientated as direct repeats and inserted into introns flanking exons that encode critical structural components of receptor[23,64]. This genetic manipulation predisposes the intervening region to deletion by Cre-recombinase[65]. We prepared slices of hippocampus from double homozygote neonatal mice (grin2a$^{fl/fl}$ grin2b$^{fl/fl}$) and cultured them in vitro for a week before co-transfecting individual CA1 neurons with cDNA for Cre-recombinase (fused in-frame with Green Fluorescent Protein, Cre-GFP) together with wildtype or mutant human GRIN2A cDNA. Slices were cultured for a further 1.5 weeks before performing recordings of evoked NMDA-mediated synaptic currents simultaneously from transfected and untransfected neurons at 15–20 DIV (Fig. 2a).

Using the experimental paradigm described above, we were able to compare the amplitudes of NMDA-EPSCs both within pairs of transfected and untransfected neurons, and the statistical interaction for the NMDA-EPSC properties between the GluN2A mutants (see Experimental procedures). Cre-GFP alone reduced the peak NMDA-EPSCs to 12% of the peak amplitude in neighbouring untransfected neurons (95% CI [9, 17]) (Fig. 2b: DKO) indicating that the NMDA-EPSCs were largely mediated by GluN2A and/or GluN2B two subunits. With the further addition of DQP-1105 (10 μM), a selective antagonist of GluN2C/D, NMDA-EPSCs were barely resolvable above baseline noise with the peak amplitudes, being 6% that of neighbouring untransfected neurons (95% CI [4, 9]) (Fig. 2b: DKO-DQP). Since most of the current was mediated by GluN2A/B, we continued with experiments to rescue NMDA-EPSCs from the GluN2A/B double knockout (DKO) neurons using mutant

human GluN2A variants. We found strong evidence in favour of an effect of expressing different GRIN2A mutants on NMDA-EPSC amplitudes (Fig. S1.1, $F_{(5,87)} = 11.4$, $p < 0.001$, $BF_{10} = 4.14 \times 10^6$). Overall, the mutants differed significantly from WT in their ability to rescue NMDA-EPSCs (Fig. 2c and Fig. S1.1, $F_{(1,87)} = 31.2$, $p < 0.001$) and we could resolve a significant difference between putative LOF (C436R, T531M, R518H) and GOF (K669N and L812M) mutations (Figs. 2c and S1.1, $F_{(1,87)} = 25.9$, $p < 0.001$). The NMDA-EPSC peak amplitudes for LOF mutants relative to WT (100%) were 35% for C436R, 38% for T531M and 37% for R518H, 95% CIs [25, 49], [25,57] and [26,53], respectively (Fig. 2c). In contrast, the K669N and L812M exhibited more effective functional rescue of the peak NMDA-EPSC, them being 64% for both K669N and L812M, 95% CIs [46,89] and [46,91], respectively (Fig. 2c). Together, these data show that the GOF GluN2A mutants were more effective (than LOF GluN2A mutants) at rescuing NMDA function at synapses in neurons largely devoid of native GluN2 expression.

The GOF mutants tested were previously reported to have much slower deactivation time constants when expressed in heterologous expression systems[57,58]. We next examined the decay time constants of the NMDA-EPSCs from the GOF mutations and compared these to WT and GluN2A and the more slowly deactivating GluN2B subunit (Figs. 2d and S1.2). Compared to WT (100%), the decay of NMDA-EPSCs in neurons rescued with K669N was 184%, 95% CI [146, 231], whereas L812M was 492%, 95% CI [386, 625], which is more like the slower NMDA-EPSC decay observed in neurons rescued with GluN2B WT (Fig. 2d). In summary, these rescue experiments broadly support the mutation classification determined by analysis of heterologous expression data and presented in Fig. 1[57,58].

**Loss-of-function GluN2A mutants vary in their ability to deliver NMDA receptors to synapses.** To test whether the poor functional incorporation of LOF mutants reflected an absence of the mutant NMDARs at synapses, we adopted a simple assay that takes advantage of the obligatory assembly of GluN1 subunits into NMDARs[27] and the ER-retention of unassembled GluN1 in neurons[66]. GFP (or its pH-sensitive variant supereccliptic pHlourin (SEP)) fused to the extracellular (or ER-luminal) amino terminus of GluN1 has previously been used to investigate trafficking competence of different GluN2 variants to synapses[27,55,67]. Exogenous expression of SEP-GluN1 alone in cultured hippocampal neurons gave largely dendritic shaft fluorescence (Fig. 3a). However, co-expression with WT GluN2A drove GluN1 into the cell membrane within dendritic spines, as evidenced by an almost 2-fold enrichment of SEP-GluN1 at spine heads colocalizing with a postsynaptic density marker, Homer1C-tdTomato. We rationalised that GluN2 mutations that disrupt the normal enrichment of NMDARs at synapses would also be incompetent at driving SEP-GluN1 to synapses. Indeed, we found strong evidence in favour of an effect of expressing different GRIN2A mutants on the spine enrichment of NMDA receptors relative to colocalized Homer1c (Fig. S1.3, $F_{(5, 185)} = 10.3$, $p < 0.001$, $BF_{10} = 4.82 \times 10^4$). Significant differences could be resolved for all orthogonal contrasts indicating that mutations differed within, as well as between, mutation types (Figs. 3 and S1.3). Synaptic levels of NMDARs in cells expressing the C436R GluN2A mutant were 56% that of WT, 95% CI [46,68], similar to that seen in neurons not cotransfected with any GluN2A (Fig. 3bii). On the other hand, GluN2A T531M and L812M exhibited intermediate levels of SEP-GluN1 spine fluorescence (75% and 74%, 95% CIs [64, 88] and [60, 92], respectively), whereas K669N efficiently rescued spine SEP-GluN1 levels to 109% of WT, 95%

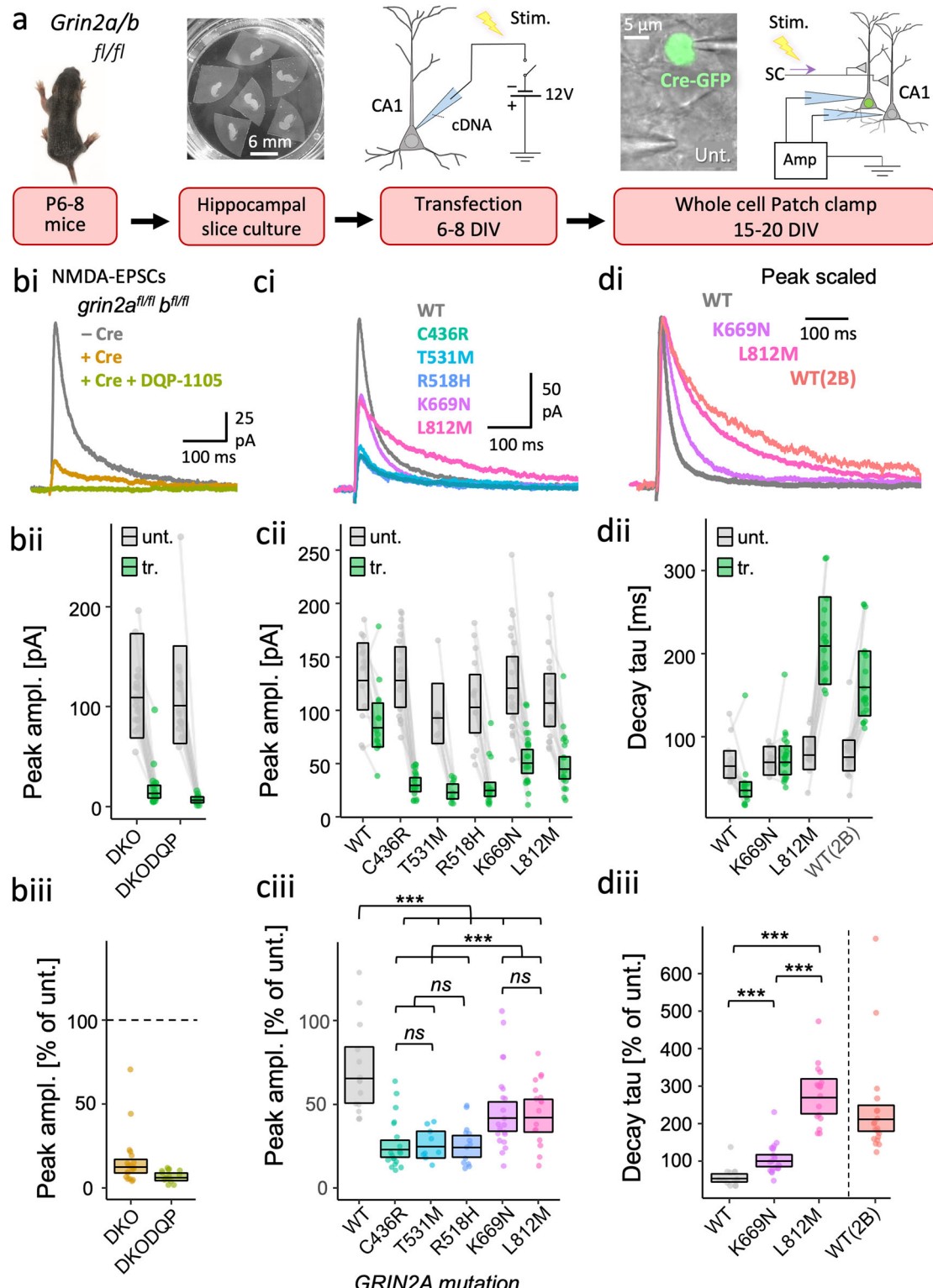

CI [90, 132]. Most striking of all, despite the poor functional rescue we observed earlier for R518H, expression of this mutant restored NMDA receptor levels in spines to 97% that of WT, 95% CI [82, 114]. Together, this suggests that R518H loss of function arises from functional defects while C436R (and to some extent T531M) exhibit defective enrichment at synapses. We note that for the L812M mutant, transfected cell density appeared very low in cell counts and was rescued by incubating the cells after transfection with NMDA receptor blocker memantine, which

suggests that overexpressing GluN2A L812M could be neurotoxic (Fig. S2). In summary, there is substantial heterogeneity in the capability of GluN2A mutants to facilitate enrichment of NMDA receptors at synapses.

**Both severe gain- and loss-of-function types of GluN2A mutant can prolong the decay of NMDA-EPSCs.** The GluN2A mutants examined above were extremely varied in their abilities to contribute functional receptors at synapses. To understand

**Fig. 2 Gain-of-function GluN2A mutants are more effective than loss-of-function GluN2A mutants at rescuing NMDAR-mediated EPSCs in GluN2A/B double knockout neurons. a** Experimental protocol used to test functional incorporation of GluN2A mutants. Plasmids expressing Cre-GFP and human *GRIN2A* cDNA were cotransfected into CA1 pyramidal neurons of organotypic hippocampal slices from *grin2a*$^{fl/fl}$ *b*$^{fl/fl}$ mice. With transfection, the action of Cre-GFP at floxed *grin2a* and *grin2b* alleles knocks-out native mouse GluN2A and 2B protein expression concurrently with expressing human GluN2A variants (wildtype or mutant). Untransfected neurons continue to express native GluN2A and B. The image of the P6 mouse pup was adapted from the 'JAX® Mice Pup Appearance by Age' poster (available from https://jackson.jax.org/) **b** NMDA-EPSC$_{+20\ mV}$ peak amplitudes from *grin2a*$^{fl/fl}$ *b*$^{fl/fl}$ (untransfected, -Cre), *grin2a*$^{-/-}$ *b*$^{-/-}$ (transfection + Cre, double KO or DKO) neurons, or DKO neurons with aCSF supplemented with selective GluN2C/D antagonist DQP-1105 (10 μM). The residual current in the DKO was effectively blocked by DQP-1105. **c** NMDA-EPSC$_{+20\ mV}$ peak amplitudes in *grin2a*$^{fl/fl}$ *b*$^{fl/fl}$ (untransfected) neurons and *grin2a*$^{-/-}$ *b*$^{-/-}$ neurons rescued with human GluN2A WT, LOF (C436R, T531M or R518H) or GOF (K669N or L812M) mutants (transfected). **d** NMDA-EPSC$_{+20\ mV}$ decay time constant (tau) in *grin2a*$^{fl/fl}$ *b*$^{fl/fl}$ (untransfected) neurons, and *grin2a*$^{-/-}$ *b*$^{-/-}$ neurons rescued with human GluN2A WT, GOF (K669N and L812M) mutants or GluN2B (transfected). **b–d** (**i**). Representative NMDA-EPSCs from transfected neurons (and untransfected neuron in (**b**) for each experimental group (**ii**) Data points of measurements made in individual neurons. Matched data points, for simultaneously recorded untransfected and transfected neurons, are connected by a line. (**iii**) Response ratios (transfected/untransfected) are expressed as a percentage and plotted for each pair of transfected-untransfected neurons. Crossbars in (**ii**) and (**iii**) show the estimated marginal means with 95% confidence intervals backtransformed from the linear mixed models (Fig. S1.1 and 1.2 for (**c**) and (**d**), respectively). Hypothesis tests in: (**iii**) are orthogonal contrasts based on a priori clustering of the mutations in Fig. 1e (Table S1); and in (**iii**) are *posthoc* pairwise comparisons using Westfall stepwise procedure. Standardised effect sizes (*r*) for comparisons of each mutant with WT for response ratios of: (**iii**) peak amplitudes were −0.55, −0.46, −0.50, −0.28 and −0.26 for mutants C436R, T531M, R518H, K669N and L812M respectively (*N* = 93); or of (**iii**) decay taus were 0.60 and 0.88 for mutants K669N and L812M, respectively (*N* = 52).

how the missense mutations are likely to affect synaptic transmission with native GluN2B expression intact, we next assessed the effect of GluN2A mutant rescue in *grin2a* knockout neurons. Experiments with WT GluN2A showed that rescue of the peak amplitude and decay time constant of NMDA-EPSCs was highly sensitive to the level of GluN2A expression (Fig. S3). The concentration of WT *GRIN2A* plasmid that effectively rescued peak amplitude and decay time of NMDAR EPSCs in GluN2A KO (3.8 ng/μL, Fig. S3) was used to assess rescue of NMDA-EPSCs for GluN2A mutants (Fig. 4). Despite mutant NMDA-EPSCs being consistently smaller than WT, we could not resolve a significant difference for peak amplitude in the contrast between mutants and WT (Fig. 4a, Fig. S1.4, $F(1, 119) = 2.47$, $p = 0.12$), and the evidence was overall strongly in favour of there being no effect among the peak amplitudes across the mutants (Fig. S1.4, $BF_{10} = 0.084$). Relative to WT (100%), estimates of the peak NMDA-EPSCs amplitudes appeared slightly smaller for LOF mutants (80% for C436R, 80% for T531M and 74% for R518H, 95% *CI*s [57, 111], [57, 111] and [53, 103] respectively), than for GOF mutants (87% for K669N and 91% for L812M, 95% *CI*s [65, 116] and [68, 122], respectively).

While examining the NMDA-EPSCs, we noticed that when scaling the NMDA-EPSCs to normalise their amplitudes, the decay kinetics appeared to differ between the genotypes (Fig. 4bi). In quantifying and comparing the time constant of the NMDA-EPSCs we found strong support for an effect of GluN2A mutation on NMDA-EPSC decay (Fig. S1.5, $F(5, 119) = 12.1$, $p < 0.001$, $BF_{10} = 3.43 \times 10^8$). Overall, the decay time constants of the mutants were significantly different from WT (Figs. 4b and S1.5, $F(1,119) = 37.1$, $p < 0.001$) as well as between GOF and LOF mutation types (Figs. 4b and S1.5, $F(1, 119) = 28.3$, $p < 0.001$). Relative to WT (100%), the decay time constants were slower for GOF mutants, it being 138% for K669N and 143% for L812M, 95% *CI*s [111, 171] and [115, 178], respectively. Surprisingly, not only was the direction of effect the same for LOF mutants, but the effect was also larger, with the decay time constants for LOF mutants relative to WT being 201% for C436R, 213% for T531M and 199% for R518H, 95% *CI*s [158, 256], [167, 272] and [156, 254], respectively.

Since we detected changes in the decay of NMDA-EPSCs, we next assessed whether there would be any effect of *GRIN2A* mutations on NMDA-EPSC charge transfer by taking the integral of the current traces (Fig. 4ci). Significant differences could be resolved in orthogonal contrasts comparing all mutants with WT

(Fig. 4c and Fig. S1.6, $F(1, 119) = 4.18$, $p = 0.04$), and between the GOF and LOF mutation types (Fig. 4c and Fig. S1.6, $F(1, 119) = 4.26$, $p = 0.04$). However, the effects were relatively small and the evidence was more in favour of there being no effect of mutation on NMDA-EPSC charge transfer (Fig. S1.6, $BF_{10} = 0.632$). Compared to WT (100%), NMDA-EPSC charge transfer was slightly larger for LOF mutations C436R (162%), T531M (177%) and R518H (126%), than it was for GOF mutations K669N (125%) and L812M (114%), 95% *CI*s [106, 247], [116, 270], [83, 193], [86, 182] and [78, 167], respectively.

Overall, the effects LOF and GOF GluN2A mutations on NMDA-EPSC decay kinetics were large and in the same direction, but because of the slightly smaller mutant NMDA-EPSC amplitudes, the effect that the slowed kinetics had on NMDA-EPSC charge transfer was much more subtle.

**Marginal effects of loss-of-function GluN2A mutants on the amplitude of AMPA-EPSCs.** NMDARs are key regulators of synaptic strength and genetic deletion of NMDAR subunits has been shown to upregulate AMPAR-mediated synaptic transmission[23,68]. To assess whether alterations to levels of synaptic AMPAR receptors could occur as a consequence of less synaptic NMDA receptors for some GluN2A mutants, we also measured evoked AMPAR-receptor-mediated EPSCs (AMPA-EPSCs). While we resolved a significant difference for AMPA-EPSC amplitude in the contrast between GOF and LOF mutations (Fig. S4 and S1.7, $F(1, 119) = 7.23$, $p = 0.008$) the effects were relatively small, and overall the evidence was more in favour there being no effect of mutation on synaptic AMPA receptors (Fig. S1.7, $BF_{10} = 0.276$). Peak amplitudes of LOF mutants relative to WT (100%) were 122% for C436R, 117% for T531M and 112% for R518H, 95% *CI*s [82, 183], [78, 175] and [75, 167], respectively. In contrast, the amplitudes of AMPA-EPSCs in neurons with GOF mutations were 86% for K669N and 85% for L812M, 95% *CI*s [60, 123] and [59, 122], respectively. The evidence here for an effect of NMDA receptor loss-of-function mutations on AMPA-EPSCs is not as convincing as the results from NMDA receptor subunit deletion experiments reported by others[23,68].

**Intermediate effects of a heterozygote *grin2a* loss-of-function allele on NMDA-EPSCs.** The data thus far raise two important questions: (1) Why do LOF mutations have the same direction of

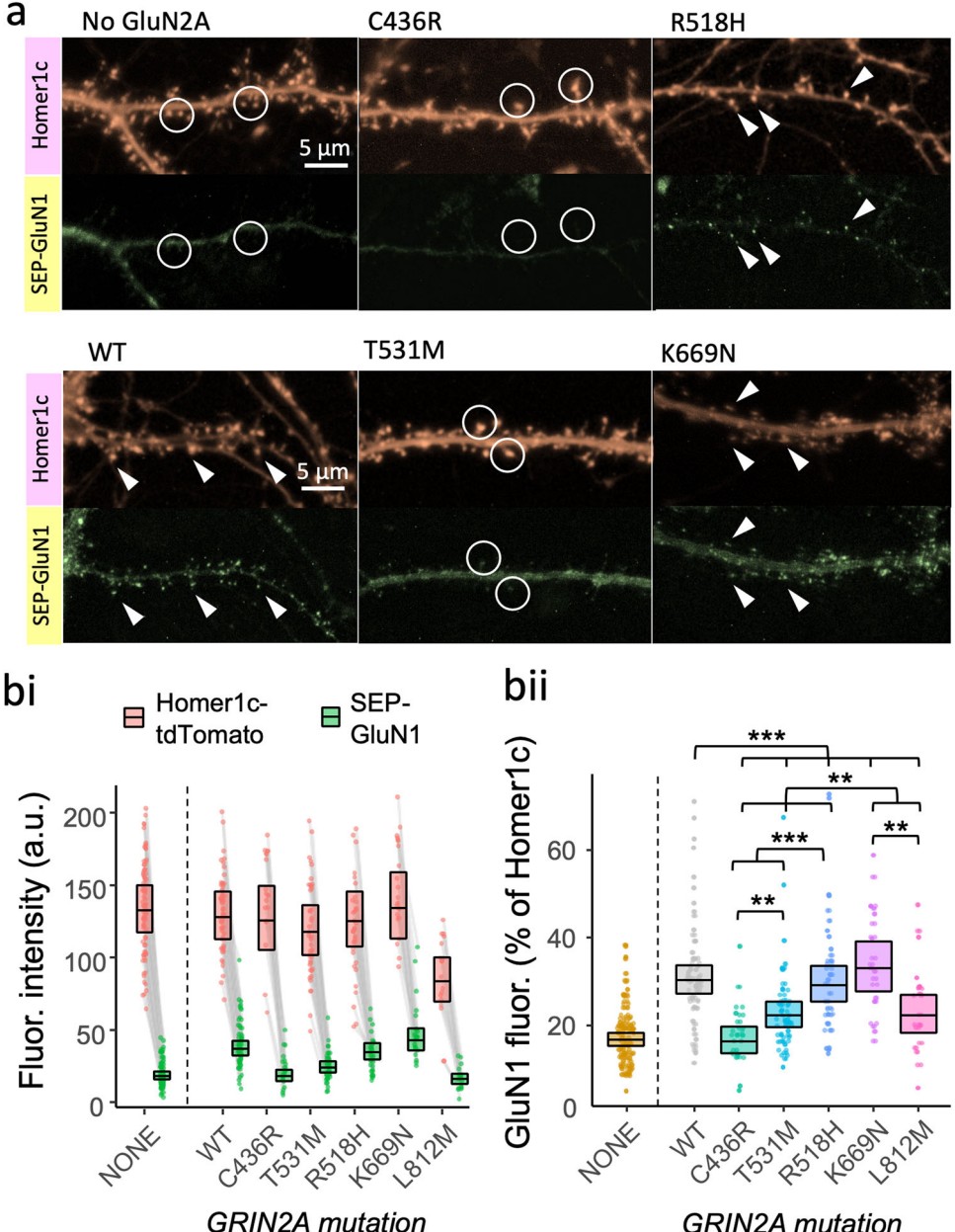

**Fig. 3 Loss-of-function GluN2A mutants vary in their ability to deliver NMDA receptors to synapses. a** Representative images of cultured neurons overexpressing GluN2A, SEP-GluN1 (green) and Homer1c-tdTomato (red). The latter was used as a synaptic marker. Arrow heads point to examples of synapses that have high synaptic levels of SEP-GluN1. Circles highlight examples of synapses that exhibit low-to-moderate levels of SEP-GluN1. **b** Levels of synaptic enrichment of NMDA receptors and Homer1c in cultured hippocampal neurons transfected with or without human GluN2A WT, LOF (C436R, T531M or R518H) or GOF (K669N or L812M) mutants. Each data point represents mean fluorescence intensity of SEP-GluN1 (green) or Homer1c-tdTomato (red) across all visible putative synaptic regions-of-interest (ROIs) in a neuron. **bi** Data points for SEP-GluN1 (green) and Homer1c-tdTomato (red) fluorescence measured from the same neuron are connected by a line. **bii** Response ratios (SEP-GluN1 / Homer1C-tdTomato) are expressed as a percentage and plotted as a point for each neuron. Crossbars in (**i**) and (**ii**) show the estimated marginal means with 95% confidence intervals backtransformed from the fitted linear mixed models (Fig. S1.3). Hypothesis tests in (**bii**) are orthogonal contrasts based on a priori clustering of the mutations in Fig. 1e (Table S1). Standardised effect sizes ($r$) for comparisons of each mutant with WT were −0.39, −0.26, −0.03, 0.06 and −0.20 for mutants C436R, T531M, R518H, K669N and L812M respectively ($N = 191$).

effect on NMDA-EPSC time course as the GOF mutations? and (2) What effect can we expect in heterozygotes for LOF alleles, especially considering that patients with *GRIN2A* mutations have only one mutant allele? We addressed these questions by examining the effect of homozygous and heterozygous null alleles on evoked NMDA-EPSCs. We achieved this genetic manipulation by transfecting Cre-recombinase in pyramidal neurons in CA1 of organotypic slices prepared from *grin2a*$^{+/+}$, *grin2a*$^{fl/+}$

and *grin2a*$^{fl/fl}$ mouse pups. We found some evidence against differing effects of Cre across the genotypes for NMDA-EPSC charge transfer (Fig. S1.8, $p = 0.30$, $BF_{10} = 0.277$), and even the apparent upward trend was not statistically significant for increasing doses of mutant allele in a linear polynomial contrast (Figs. 4a and S1.8, $F(1,96) = 2.39$, $p = 0.13$). However, a differing effect of Cre across the genotypes was strongly supported for NMDA-EPSC peak amplitude, decay time constant and rise-time

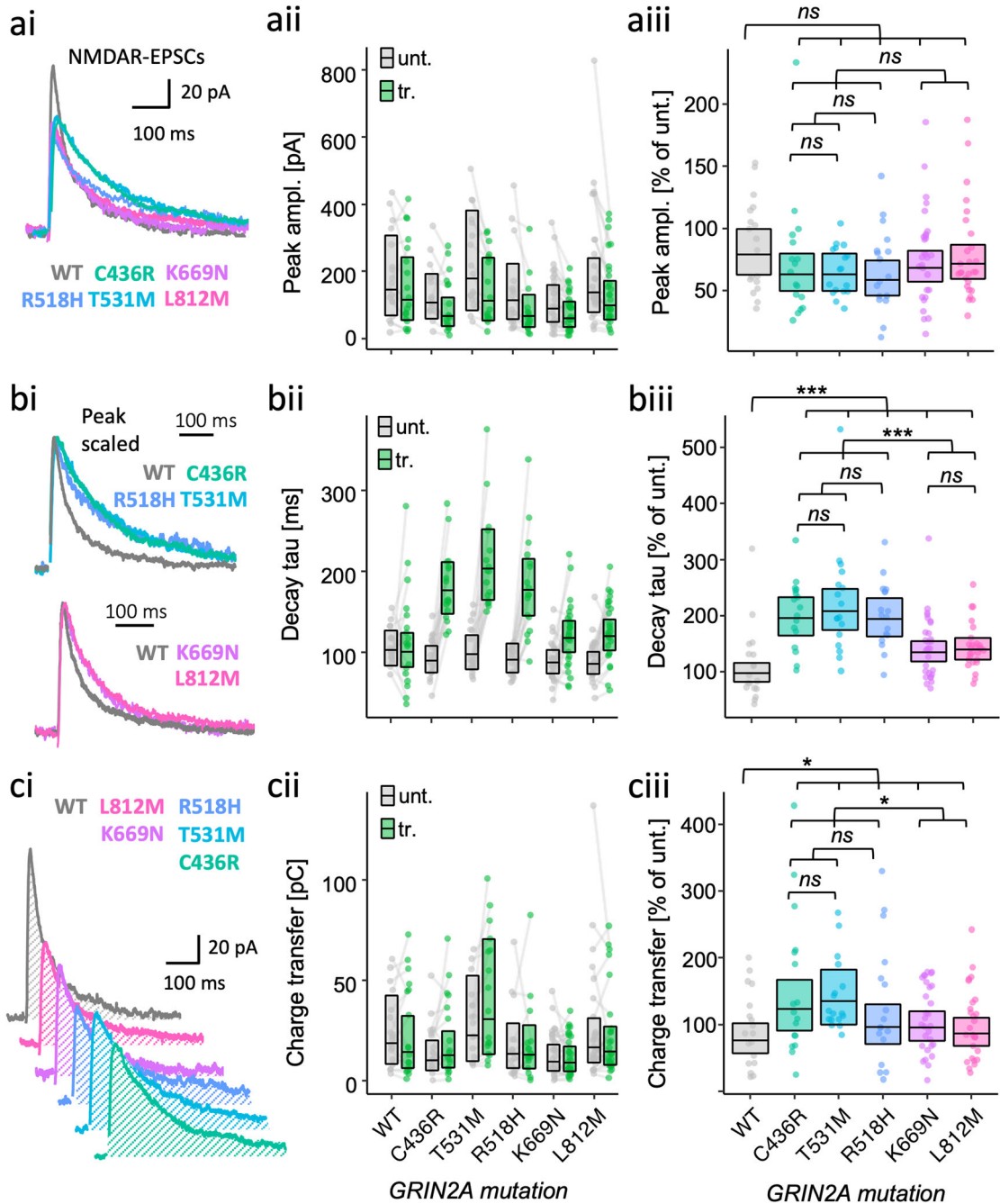

**Fig. 4 Gain- and loss-of-function LBD mutants are both associated with more prolonged NMDAR-mediated EPSCs.** NMDA-EPSC$_{+20\ mV}$ peak amplitudes (**a**), decay time constants (**b**), or charge transfer (**c**) in *grin2a$^{fl/fl}$* (untransfected) neurons, and *grin2a$^{-/-}$* neurons rescued with human GluN2A WT, LOF (C436R, T531M or R518H) or GOF (K669N or L812M) mutants (transfected). **a–c** (**i**). Representative NMDA-EPSCs from transfected neurons for each experimental group. (**ii**) Data points of measurements made in individual neurons. Matched data points, for simultaneously recorded untransfected and transfected neurons, are connected by a line. **iii** Response ratios (transfected/untransfected) are expressed as a percentage and plotted for each pair of transfected-untransfected neurons. Crossbars in (**ii**) and (**iii**) show the estimated marginal means with 95% confidence intervals backtransformed from the linear mixed models (Fig. S1.4–S1.6). Hypothesis tests are orthogonal contrasts based on a priori clustering of the mutations in Fig. 1e (and Table S1). Standardised effect sizes (*r*) for comparisons of each mutant with WT for response ratios of: (**aiii**) peak amplitudes were −0.12, −0.12, −0.16, -0.09 and -0.06; (**biii**) decay taus were 0.46, 0.49, 0.46, 0.26 and 0.28; and (**ciii**) charge transfer were 0.20, 0.24, 0.10, 0.11 and 0.06, for mutants C436R, T531M, R518H, K669N and L812M respectively (*N* = 125).

(Figs. S1.9–1.11, $BF_{10}$ = 227, 4.67 × 10$^{19}$ and 2.33 × 10$^{13}$ respectively, *p* < 0.001 for each). Significant differences could be resolved for all pairwise comparisons between *grin2a$^{+/+}$*, *grin2a$^{-/+}$* and *grin2a$^{-/-}$*, except for the comparison of peak amplitudes between *grin2a$^{+/+}$* and *grin2a$^{-/+}$* (Fig. 5c, d, and S1.9-1.11). Consistent with the NMDA-EPSCs observed in the

rescue experiments of the LOF GluN2A mutants, the decay time constant was 207% slower for *grin2a$^{-/-}$* (95% *CI* [183, 233]) relative to WT g*rin2a$^{+/+}$*, 100%). Together with the negligible NMDA-ESPC in the double knockout (*grin2a$^{-/-}$ grin2b$^{-/-}$*, Fig. 2b), this suggests that the residual current is mediated by GluN2B, which has slower deactivation kinetics and could explain

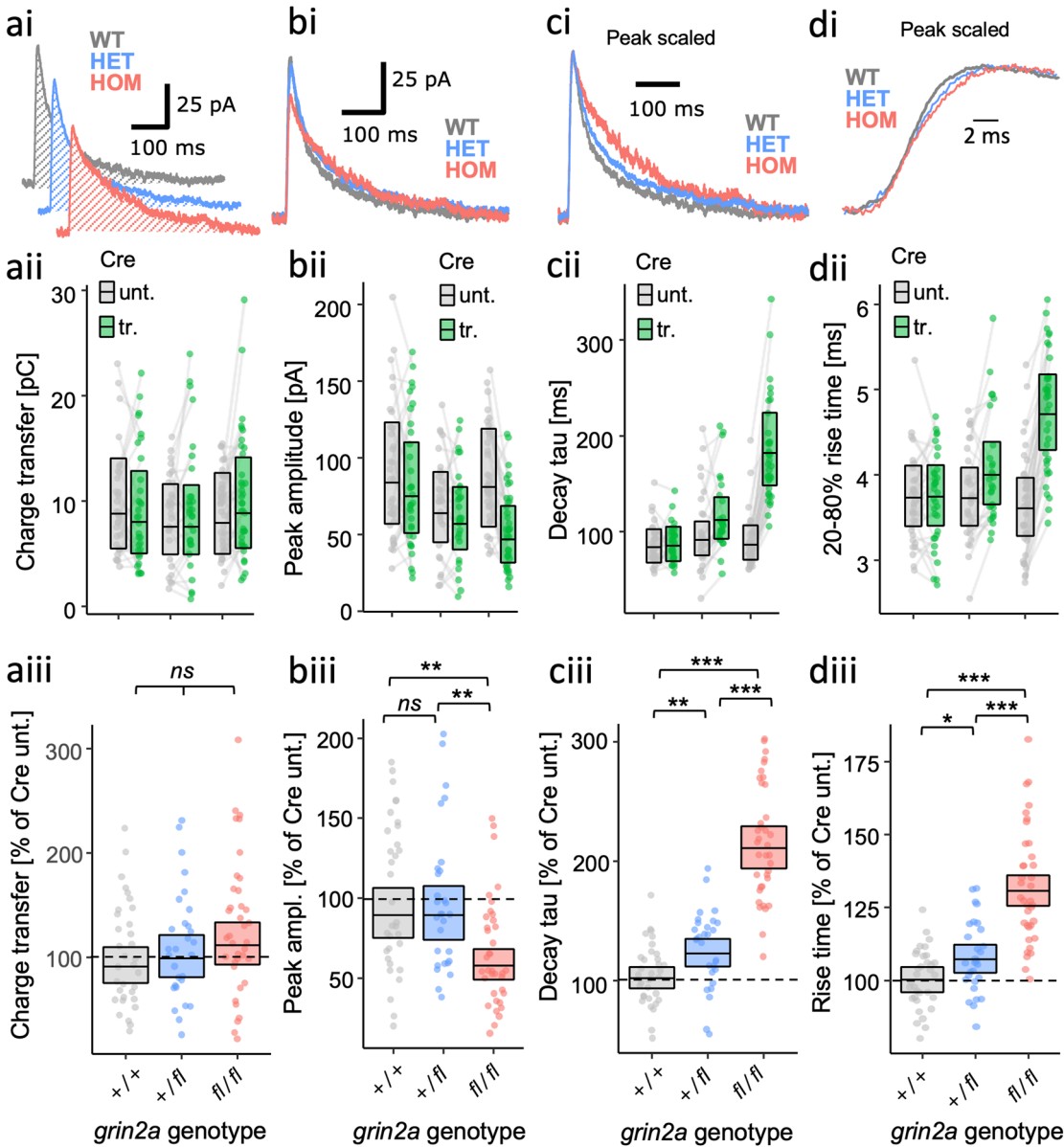

**Fig. 5 Dose-dependent effects of *grin2a* null alleles on NMDA-EPSC kinetics.** NMDA-EPSC$_{+20\ mV}$ charge transfer (**a**), peak amplitude (**b**), decay time constant (**c**), or 20-80% rise-time (**d**) in *grin2a*$^{+/+}$, *grin2a*$^{+/fl}$ or *grin2a*$^{fl/fl}$ (untransfected) neurons and *grin2a*$^{+/+}$, *grin2a*$^{+/-}$ or *grin2a*$^{-/-}$ neurons (transfected, with Cre-GFP). **a–d** (**i**). Representative NMDA-EPSCs from transfected neurons for each experimental group. (**ii**) Data points of measurements made in individual neurons. Matched data points, for simultaneously recorded untransfected and transfected neurons, are connected by a line. (**iii**) Response ratios (transfected/untransfected) are expressed as a percentage and plotted for each pair of transfected-untransfected neurons. Crossbars in (**i**) and (**ii**) show the estimated marginal means with 95% confidence intervals backtransformed from the fitted linear mixed models (Figs. S1.8–S1.11). Hypothesis tests are *posthoc* pairwise comparisons using Westfall stepwise procedure. Standardised effect sizes (*r*) for comparisons of each genotype with WT for response ratios of: (**aiii**) charge transfer was 0.06 and 0.16; (**biii**) peaks amplitudes were 0.00 and −0.35; (**ciii**) decay taus were 0.28 and 0.77; and (**diii**) rise-time were 0.22 and 0.68, for heterozygous and homozygous genotypes respectively (*N* = 99).

the more prolonged NMDA-EPSCs observed also for LOF GluN2A mutants (Fig. 4b). The effect observed for NMDA-EPSC decays in *grin2a*$^{-/+}$ neurons (i.e., heterozygotes) was 120% (95% *CI* [106, 137]), it being intermediate between wildtype and homozygous cases. Overall, these results illustrate that even a single null allele of *grin2a* is sufficient to cause defects in synaptic transmission.

**Mutation-associated changes in NMDA-EPSCs can shape synaptic excitability and calcium dynamics.** The results raise questions about how the changes in NMDA-EPSCs by the mutations will ultimately affect excitability. In particular, how could smaller and slower NMDA-EPSCs alter excitability when

the amount of charge transfer is effectively unchanged? Using a computational model of a CA1 pyramidal neuron, complete with a wide range of active conductances[69] and synaptic AMPA and NMDA receptors[70] (Fig. 6ai), we simulated the effects of slowing the kinetics of NMDA-EPSCs to 150% and 200% compared to WT (100%), which roughly correspond to the effects observed for GOF and LOF mutants respectively. In each case the synaptic weight (a.k.a peak amplitude) was also scaled to equalise charge transfer for single EPSCs (Fig. 6aii and iii). We then simulated short trains of synchronous synaptic activation and compared spine head currents and intracellular calcium levels at activated synapses between the different NMDA-EPSC profiles.

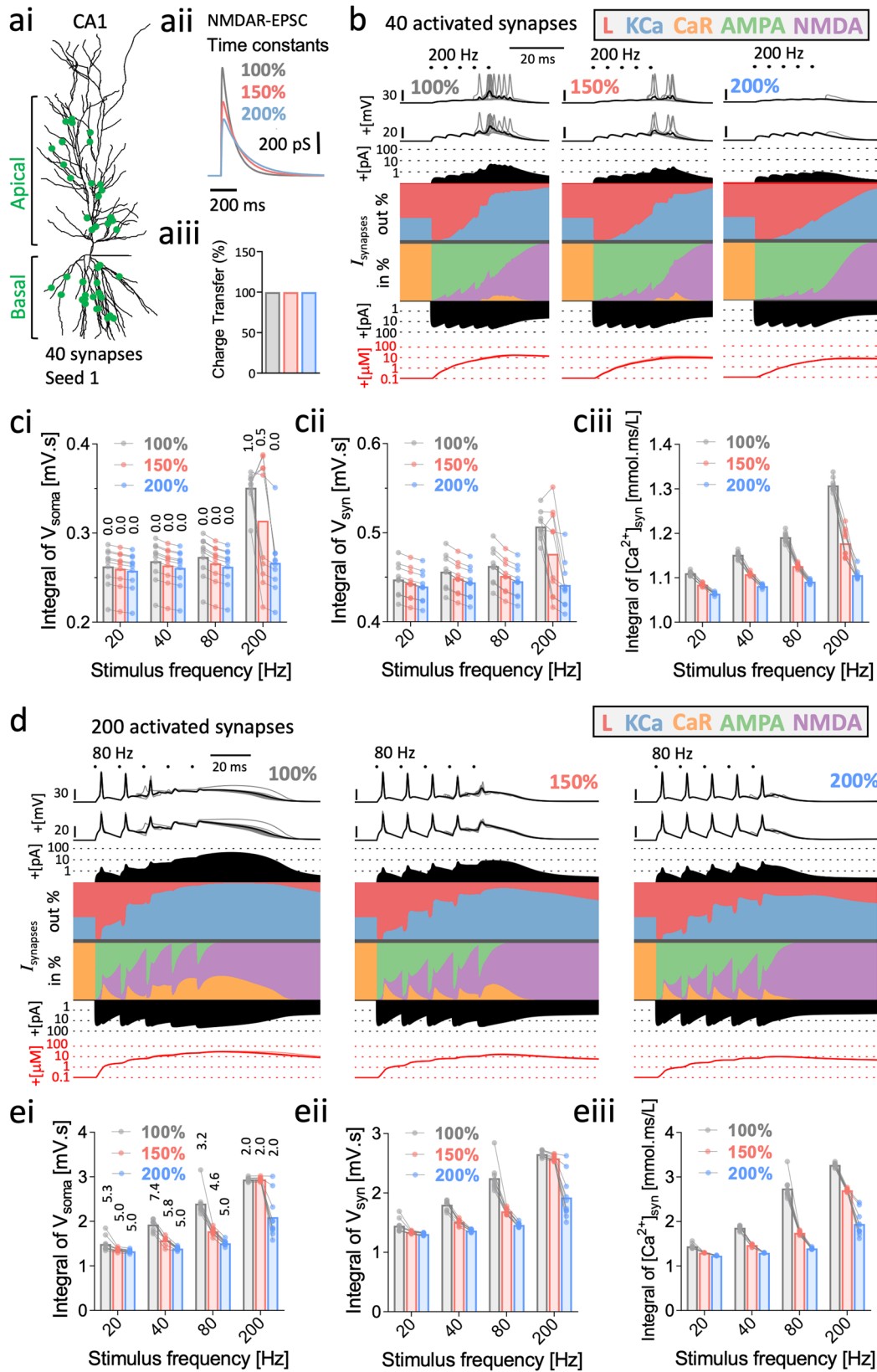

Trains of 40 randomly positioned synapses, simultaneously activated over a range of frequencies (20, 40, 80 and 200 Hz) were simulated and the membrane potential (in soma and spine heads) and inward currents, outward currents and calcium concentrations in spine heads were stored, averaged across spines and summarised as current scapes[71] (Fig. 6b). The simulation was

repeated with 10 different random seeds for each NMDA-EPSC profile and stimulation protocol. While an individual EPSP burst was subthreshold across the three groups, WT-like (100%) EPSP trains drove neuronal spiking by the 4th or 5th EPSP during short high frequency (200 Hz) trains, and the action potentials backpropagated to spine heads (Fig. 6b, ci). The voltage changes

**Fig. 6 Spine calcium dynamics, temporal summation, and spiking output during EPSP trains are shaped by mutation-associated changes in NMDA-EPSCs. ai** Model of the CA1 pyramidal neuron[69] used for the simulations. **aii** The kinetics of the NMDAR-component of the synaptic conductance was slowed by 150% and 200% and the amplitude of the NMDAR-component was reduced to equalise charge transfer to mimic the effects of GOF (red) and LOF (blue) mutants. **aiii** Bar graph illustrating the constant charge transfer during the EPSCs of varying kinetics to mimic GOF and LOF mutants. **b** Current scapes for 40 activated excitatory synapses stimulated 5 times at 200 Hz. The NMDA component of the synaptic conductance had WT-like (100%), GOF-like (150%), or LOF-like (200%) kinetics with synaptic weight compensated to normalise charge transfer. From top-to-bottom, current scapes show the membrane potential at the soma and ensemble mean membrane potential, outward current (amplitude and %), inward current (% and amplitude) and intracellular calcium concentration in spine heads at activated synapses. **c** Data points overlaying bar charts of mean integral of soma membrane potential (**i**) and spine head membrane potential (**ii**) or intracellular calcium (**iii**) for 40 excitatory synapses stimulated 5 times at 20, 40, 80 and 200 Hz. Lines connecting the points reflect data simulated with the same random seed ($n = 10$). Numbers labelling bars in (**i**) represent mean number of action potentials per train. **d** Current scapes as in (**b**) but for 200 activated excitatory synapses stimulated 5 times at 80 Hz. **e** As in (**c**) but for 200 activated excitatory synapses.

in spine heads were associated with robust activation of NMDARs, R-type $Ca^{2+}$ channels (CaR) and $Ca^{2+}$-activated $K^+$ channels (KCa) (Fig. 6b, cii) and large increases in $Ca^{2+}$ reaching a peak of >10 μM, and an integral of 1.3 mmol.ms/L (Fig. 6b, ciii). At the same frequency, neuronal spiking was either lower or absent for the GOF-like (150%) or LOF-like (200%) NMDA-EPSP condition respectively (Fig. 6b, ci). These changes in spiking were also associated with less activation of NMDARs, CaR and KCa, and less accumulation of $Ca^{2+}$, in spines for mutant-like NMDA-EPSC profiles (Fig. 6b, ciii). Summation of EPSPs at slower frequencies did not result in any spiking and the differences between NMDA-EPSC profiles for depolarisation and intracellular $Ca^{2+}$ became smaller (Fig. 6ciii).

The full set of simulations above were repeated with 200 synchronous synaptic inputs to evaluate the effect of NMDA-EPSC profiles on the summation of suprathreshold EPSPs. WT-like (100%) (but not mutant-like) EPSP trains at 80 Hz and above triggered plateau potentials that were associated with robust activation of spine NMDARs, CaR and KCa (Fig. 6d). Interestingly, the spiking output was very frequency dependent, with there being greater spiking in mutant-like NMDA-EPSCs at 80 Hz, but less spiking at 40 Hz and similar spiking at the more extreme train frequencies (Fig. 6d, ei). Examination of the voltage traces at 80 Hz revealed that spiking was strongly attenuated during the plateau potential, likely due to Na-channel inactivation. Overall spine $Ca^{2+}$ accumulation reached up to 2–3 times the levels compared to 40 active synaptic inputs, and the mutant-like NMDA-EPSC profiles still lead to less accumulation of spine $Ca^{2+}$ (Fig. 6eiii).

Overall, these results indicate that even in the absence of changes in charge transferred during synaptic activation, mutant NMDAR kinetics could still effectively modulate synaptic excitability and spine calcium dynamics, with the effects being very frequency dependent.

## Discussion

Our investigation revealed how mutations from patients with *GRIN2A*-related epilepsy disorders result in defective synaptic transmission. The robust differences in the ability of mutant GluN2 subunits to rescue NMDA-EPSCs in GluN2-null (grin2a$^{-/-}$ b$^{-/-}$) neurons (Fig. 2) were largely consistent with the classification of mutations, as severe GOF and LOF, based on analysis of data obtained in heterologous expression systems by others (Fig. 1). Remarkably though, replacing only the native GluN2A with the mutant receptor subunit gave rise to NMDA-EPSCs with very similar defects for LOF and GOF mutations: NMDA-EPSCs that were slower in time course but similar in charge transfer (Fig. 4). In general, mutations phenocopied the GluN2A knock-out (Fig. 5), but the mechanism underlying the defect varied, with some mutants being associated with poor trafficking to synapses (Figs. 3 and 7b) while others reached the synapse but

were either functionally silent (Figs. 2, 3, 7c) or contributed directly to the NMDA-EPSCs through altered functional properties of the mutant receptor (Figs. 2, 7d). In the case of the LOF mutations, the changes in the NMDA-EPSC time course likely reflect the contribution of natively expressed GluN2B (Figs. 5, 7b, c).

The ability of GluN2A mutants to incorporate into synaptic NMDARs varied considerably (Figs. 2, 3). The LOF mutation C436R gave similarly low levels of synaptic enrichment of NMDARs as the control without GluN2A (Fig. 3). The C436R mutation disrupts the formation of a disulfide bridge, which is likely key to the folding and stability of the GluN2A protein structure (Fig. S5). In contrast, the LOF R518H gave similar levels of NMDARs at synapses as wildtype GluN2A (Fig. 3), and the lack of functional incorporation of R518H (Fig. 2c) may be explained by the effect of the mutation on lowering glutamate potency (Fig. 1c–e) by breaking coulombic interactions between GluN2A and the glutamate agonist (Fig. S5). Some mutations, like the T531M mutation, may even show a combination of these mechanisms (Figs. 2, 3 and S5). It is important to point out that some mutations may have effects not represented by this sample, the LOF mutants of which were all from the LBD. It is conceivable that mutations in other parts of the receptor could give receptors that traffic to the synapse but whose functional defects lead to distinct synaptic phenotypes. Nonetheless, our results show that mutations with severe and contrasting LOF and GOF effects on GluN2A-containing receptors can lead to effects on synaptic function that while abnormal, are in the same direction.

A current question in the field of *GRIN2A*-related disorders is whether expression or synaptic delivery of the native GluN2B subunit is specifically upregulated and compensates for GluN2A loss of function[43]. The smaller NMDA-EPSCs in neurons that are homozygous for the null GluN2A allele indicate that compensation by GluN2B could be incomplete (Figs. 5a, 2d). However, in neurons heterozygous for a null allele, the slower, GluN2B-like time course of NMDA-EPSCs in the absence of changes in amplitude suggests that there could be some, albeit small, compensation. The mechanism of this compensation may simply reflect a greater propensity for existing GluN2B-containing NMDARs to accumulate at synapses when synaptic content of GluN2A is reduced. This means of compensation would still be consistent with GluN2A deletion not having any effect on either GluN2B total protein or the amplitude of ifenprodil-sensitive whole-cell currents[43]. It should be mentioned though that since GluN2A and GluN2B subunits couple to distinct downstream signalling pathways through their distinct intracellular C-tails[11,72], changes in the GluN2A/B subunit composition at synapses could be expected to have an impact on long-term regulation of synaptic function (e.g. synaptic plasticity). As such, more complex aspects of synapse phenotype between GOF and LOF mutants may indeed differ in more than ways we can

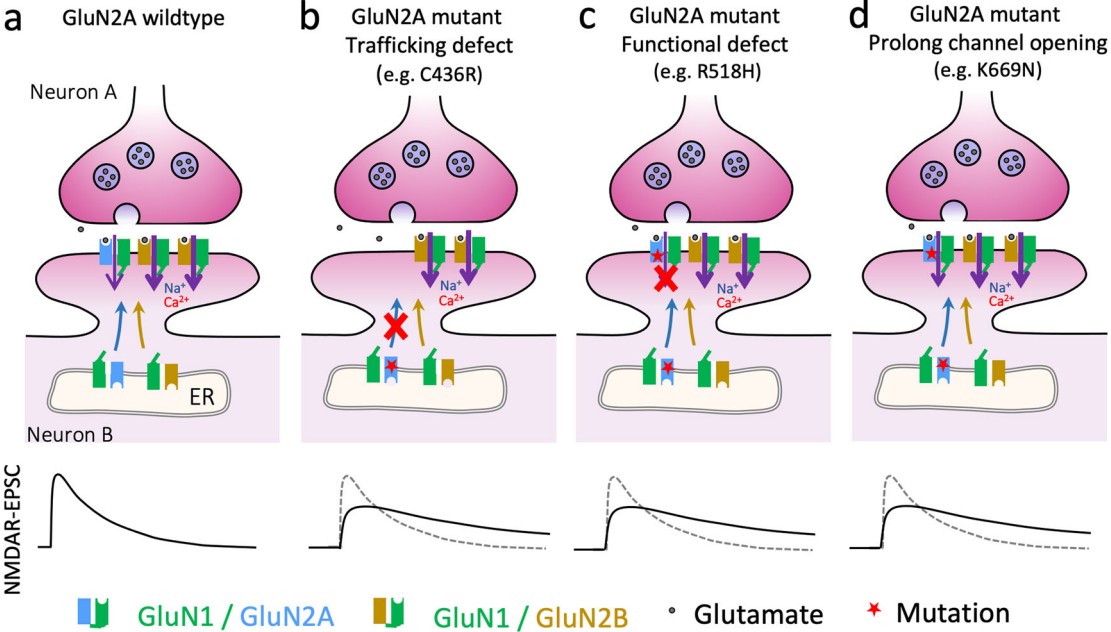

**Fig. 7 Model summarising how different molecular defects can converge on similar NMDAR-mediated EPSCs. a** WT GluN2A and GluN2B mostly contribute to NMDA-EPSCs in CA1 pyramidal neurons. GluN2B-containing NMDARs have more prolonged time course and cation influx as illustrated by the bold purple arrows. **b** GluN2A mutants with defects culminating in low levels of GluN2A at synapses result in NMDA-EPSCs that are dominated by residual GluN2B-containing NMDARs. **c** GluN2A mutants that traffic to the synapse but are functionally defective result in NMDA-EPSCs that are dominated by residual GluN2B-containing NMDARs. **d** GluN2A mutants that traffic to the synapse but have slower deactivation kinetics directly contribute to prolonging NMDA-EPSCs. **b–d** The effect of mutations may result in a combination of these different mechanisms.

determine from measuring basal synaptic transmission. While unlikely in our single-cell genetic manipulations, we also cannot rule out the possibility that disturbances in NMDA receptor function lead to downstream changes in the relative expression of GluN1 splice variants, which could also explain changes in NMDA-EPSC time course[73,74]. Although it is also possible that manipulations of postsynaptic GluN2 NMDA receptor subunits could change presynaptic release probability and interfere with interpreting effects of GluN2A mutations on NMDA-EPSC amplitude, the lack of effect of single-cell KO of NMDA receptor subunits on paired-pulse facilitation at Schaffer collateral synapses makes this seem unlikely[23].

In general, we were surprised by the relatively modest effects of GluN2A mutations on synaptic NMDA-EPSCs in our $grin2a^{-/-}$ rescue experiments (Fig. 4) compared to our results in the GluN2A/B double knockout (Fig. 2). Together with the association of these mutations with seizure disorders (Table S2), this highlights just how sensitive neuronal excitability is to the intrinsic properties and abundance of synaptic NMDARs. The actual impact of *GRIN2A* mutations on NMDA-EPSCs may even be overestimated here since we performed our rescue experiments in $grin2a^{-/-}$ (homozygote) neurons, with the risk being that the effects we observed may not be truly representative of what occurs in the neurons of patients that are heterozygous for the mutations. This could be of particular concern for LOF mutations, in which the remaining wildtype GluN2A allele could compensate for the mutant allele. Our experiment measuring NMDA-EPSCs in neurons that were heterozygous for a *grin2a* null allele addressed this concern (Fig. 5) and suggests that qualitatively, our inferences throughout could hold in the heterozygote case, although we cannot rule out the possibility that heterozygosity for GOF mutations could lead to unpredictable effects on NMDA-EPSCs.

The overall effects of GluN2A mutations on the properties of the NMDA-EPSCs raised some interesting questions about how

they may modify synaptic excitability. In particular, the slower time course and smaller amplitudes of the NMDA-EPSCs in $grin2a^{-/-}$ neurons rescued with GluN2A mutants resulted in charge transfer through the NMDAR channels being similar to wildtype. This prompted us to model the effect of the mutations on NMDA-EPSCs and simulate the depolarisations and spine calcium dynamics during trains of concerted synaptic activity onto CA1 neurons (Fig. 6). These simulations serve to illustrate that the slower time course of NMDA-EPSCs mediated by GluN2A mutants can affect EPSP summation even when charge transfer is the same. However, there may be other effects of the GluN2A that we have not measured or modelled (e.g., presynaptic or extrasynaptic receptors, synaptic plasticity) that contribute to synaptic excitability and neuronal signalling. In addition, GluN2A is expressed in other neuronal cell types (including inhibitory neurons[75]) and how the mutation effects in all cells sum together to affect excitatory-inhibitory balance and network activity is unknown. Together, understanding how NMDAR mutations affect excitability and network stability warrants further investigation and may ultimately help in designing therapeutic strategies to treating *GRIN2A*-related disorders.

In summary, severe loss- and gain-of-function *GRIN2A* mutations associated with epilepsy-aphasia spectrum disorders lead to similar dysfunctional NMDAR-mediated synaptic currents in CA1 pyramidal neurons when expressed in cultured mouse hippocampal slices, albeit via different mechanisms. Modelling suggests that mutant-like NMDA-EPSCs can lead to abnormal temporal summation and spine calcium dynamics.

## Methods

**Mutant classification**. The functional consequences of 20 *GRIN2A* mutations were collected from publications and databases[45,47,51–58]. The functional properties were Glutamate $EC_{50}$ and Glycine $EC_{50}$ (in oocytes), and deactivation time constant, current density, open probability, and relative cell surface levels (in HEK-

293T cells). Potency for R518H was approximated from the fold change in potency for R518C documented in the CFERV database[52]. Potency for T531M was approximated by scaling the R518H potency according to the R518H/T531M ratio for single-channel opening frequency corrected for the differences in surface expression[57]. To handle other remaining instances of missing data, single imputation was achieved by linear interpolation following orthogonal regression on the rank-transformed data using principal component analysis (PCA) by the alternating least-squares algorithm in Matlab (Mathworks). The complete dataset was then normalised to the wildtype (WT) GluN2A measurement for each property and transformed to the log scale. The dataset had a Kaiser–Meyer–Olkin test statistic of >0.5 and was considered suitable for PCA. More than 85% of the variance was explained by the first two principal components, which were then extracted from the loadings matrix and subjected to oblique (Varimax) rotation. The resulting pattern matrix was used to calculate rotated component scores. The number of clusters chosen for K-means clustering of the scores was determined from the elbow in a plot of the sum-of-squares error against the cluster number.

**Molecular biology**. Mutagenesis was performed on the cDNA of human *GRIN2A* in the pCI-Neo vector (pCMV GluN2A). GluN2A mutations with respect to the amino acid numbering in NP_000824.1 were C436R, T531M, R518H, K669N, L812M. Oligonucleotide primers (Integrated DNA Technologies) were designed to incorporate single-base substitutions using the GeneArt® Site-Directed Mutagenesis Kit (Thermofisher Scientific) (Table S3). The presence of each mutation was confirmed by DNA sequencing (Eurofins Genomics) and the whole coding sequence was also examined to ensure that no other mutations were introduced by polymerase chain reaction (PCR) errors. Plasmid DNA was amplified and purified using EZNA endo-free plasmid maxi kit (VWR International).

**Animals**. C57Bl/6J mice and Sprague Dawley rats were purchased from Charles River Laboratories. Rodents were maintained under conventional housing conditions and a 12–12 light–dark cycle. Heterozygous *grin2a*-flox (*grin2a*[+/fl]) and *grin2b*-flox (*grin2b*[+/fl]) mice were a gift from Kenji Sakimura (Niigata University, Japan). Generation of these mice has been described previously[23,64]. Homozygous *grin2a*-flox (*grin2a*[fl/fl]) and *grin2b*-flox (*grin2b*[fl/fl]) mice were generated by mating the respective heterozygous mice and the colonies were subsequently maintained as homozygous lines taking care to avoid brother-sister matings. Double-flox (*grin2a*[fl/fl] *2b*[fl/fl]) mice were generated by crossing the *grin2a*[fl/fl] and *grin2b*[fl/fl] mouse lines. Ear biopsies taken for identification were used for genotyping by PCR. Primers used for *grin2a*-flox genotyping were: 5′-GAAATGTGGTAAAATCC AGTTAG-3′ (forward), and 5′-TAGGCAGTAAACTTTCCTCATC-3′ (reverse). Product sizes were 800 base pairs (bp) for the WT allele and 1100 bp for the *grin2a*-flox allele. Primers used for *grin2b*-flox genotyping were: 5′-GTCAGAGA GTTTAACCTCAG-3′ (forward), 5′-ACAAGCACCTTCTTGGTCTC-3′ (reverse) and 5′-GCTGCTAAAGCGCATGCTCC-3′ (Neo). Product sizes were 940 bp for the WT allele and 200 bp for the *grin2b*-flox allele. Neonatal animals used for experiments were culled by an appropriate Schedule 1 procedure (cervical dislocation) in accordance with the Animals Scientific Procedures Act 1986 amendment regulations 2012. Schedule 1 training of the experimenters was overseen and approved by the local Named Training and Competency Officer (NTCO). Experiments were reviewed a priori by the Animal Welfare Ethical Review Body at the University of Sussex and the project was licenced by the Home Office (UK, PPL# P6CF775B9).

**Cultures**. Organotypic hippocampal slice cultures were prepared from neonatal mice. The hippocampi were dissected from mice at postnatal age 6-8 days in cold sterile filtered dissection medium composed of (in mM): $CaCl_2$ (0.5), KCl (2.5), $KH_2PO_4$ (0.66), $MgCl_2$ (2), $MgSO_4$ (0.28), NaCl (50), $Na_2PO_4$ (0.85), glucose (25), $NaHCO_3$ (2.7), sucrose (175) and HEPES (2) (pH 7.3, 330 mOsm). Transverse hippocampal slices (350 μm) were cut using a McIlwain tissue chopper (Abbotsbury Engineering Ltd., UK), placed on pieces of membrane (FHLC01300, Millipore, UK) placed in Millicell cell culture inserts (PICM03050, Millipore, UK) and subcultured in 6-well plates. The culture medium (1 ml per well) was composed of Minimum Essential Medium (MEM) supplemented with 15% heat-inactivated horse serum (26050-088, Gibco), B27 supplement (17504044, Gibco), 25 mM HEPES, 3 mM L-glutamine, 1 mM $CaCl_2$, 1 mM $MgSO_4$, 0.25 mM ascorbic acid and 5 g/L glucose. Slice culture plates were kept in an incubator at 34 °C with 5% $CO_2$ and 95% humidity. The culture medium was changed twice a week, with prewarmed slice culture medium supplemented with 10 μg/ml gentamicin.

Dissociated cultures of hippocampal neurons were prepared from neonatal Sprague Dawley rats. The hippocampus was dissected on ice and the meninges were removed in dissection medium composed of HBSS (14155048, ThermoFisher Scientific) supplemented with 10 mM HEPES (H0887, Sigma) and 100 U/ml Penicillin and Streptomycin (Pen-Strep, 15140122, ThermoFisher Scientific). The hippocampi were then triturated in 2 ml of prewarmed plating medium, which was composed of MEM (51200087, ThermoFisher Scientific) supplemented with 20 mM glucose, 100 U/ml Pen-Strep, 1 mM sodium pyruvate (Sigma S8636), 25 mM HEPES, N2 supplement (17502048, ThermoFisher Scientific) and 10% heat-inactivated horse serum (26050088, ThermoFisher Scientific). The homogenised tissue was then diluted with plating medium to 3.25 ml per

hippocampus. 500 μl of this was plated per well into 12-well plates that contained 16 mm diameter coverslips (10755354, ThermoFisher Scientific) that had been previously coated with 50 μg/ml poly-D-lysine (P0899, Sigma) and 2 μg/ml laminin (L2020, Sigma). Neurons were allowed to settle and adhere for 3 h at 37 °C, 5% $CO_2$, after which, they were supplemented with 1.5 ml culture medium containing Neurobasal-A (12349015, ThermoFisher Scientific), B-27 supplement (17504044, ThermoFisher Scientific), 100 U/ml Pen-Strep and GlutaMAX (35050038, ThermoFisher Scientific). Cultures were returned to 37 °C, 5% $CO_2$ and fed twice a week with culture medium supplemented with antimitotic agents: 0.1 μM uridine, 0.1 μM fluorodeoxyuridine and 0.1 μM cytosine arabinoside (all from Sigma).

**Transfections**. Neurons in organotypic slice cultures were transfected by single-cell electroporation[76]. At 6-8 days in vitro (DIV) CA1 pyramidal neurons in organotypic slices were electroporated with plasmids expressing pCMV Cre-GFP with or without WT or mutant pCMV *GRIN2A* cDNA (C436R, T531M, R518H, K669N, or L812M). Plasmid DNA was precipitated upon the addition of 0.5 volume of PEG solution (30% PEG 8000 (w/v) in 30 mM $MgCl_2$ to isolate it from residual lipopolysaccharide endotoxin. The DNA precipitate was pelleted by centrifugation at 10,000 relative centrifugal force (rcf) for 15 min, washed with 70% ethanol and the air-dried pellet was re-dissolved in TE buffer (10 mM Tris-HCl, 1 mM EDTA). Within a week of transfection, pCMV Cre-GFP was diluted to 0.53 nM (3 ng/μL) in intracellular solution containing (in mM): $CH_3SO_3H$ (135), KOH (135), NaCl (4), $MgCl_2$ (2), HEPES (10), Na-ATP (2), Na-GTP (0.3), spermine dihydrate (0.15), EGTA (0.06) and $CaCl_2$ (0.01) (pH 7.25, 285 mOsm). In molecular replacement experiments, the solution also included 0.62 nM (3.8 ng/μL) of WT or mutant pCMV *GRIN2A*. DNA-containing intracellular solution was centrifuged at >10,000 rcf for 15 min at 4 °C to remove debris before using it to fill patch pipettes (8–10 MΩ), which were pulled with a Flaming/Brown Micropipette puller from thick-walled borosilicate glass capillaries (GB150F-8P, Science Products). Slices were transferred to the recording chamber on an upright microscope (BX51, Olympus) containing room temperature extracellular solution composed of (in mM): NaCl (140), KCl (3), $MgCl_2$ (1), $CaCl_2$ (2), glucose (10), Na-pyruvate (1), $NaHCO_3$ (2), HEPES (6), Na-HEPES (4) (pH 7.35, 300 mOsm). The patch pipette was positioned in the slice under visual guidance using a mechanical manipulator (PatchStar, Scientifica). CA1 pyramidal cells were approached with positive pressure (20 mbar). Upon dimple formation, the pressure was released to form a putative loose-patch seal. Immediately, a 12 V stimulus train was applied (100–200 Hz for 0.25–0.5 s; pulse-width 0.25–0.5 ms) from a stimulus isolator (IsoFlex, A.M.P.I.) triggered from ACQ4 software (v0.9.3)[77] through a USB-X Series Multifunctional DAQ interface (NI USB-6341, National Instruments). Using the same patch-pipette, ten CA1 pyramidal neuron transfections were attempted in each slice. Typically, expression (judged from the fluorescent protein marker) was observed in half the neurons in which transfection was attempted.

Cultures of dissociated hippocampal neurons were transfected at 8 DIV using the calcium phosphate method[78]. DNA constructs cotransfected (μg/coverslip) were pCaMKII-α Homer1c-tdTomato (1), empty pcDNA3.1 + (1.5), and pCMV constructs driving expression of SEP-GluN1 (1.5) (Addgene plasmid #23999) and GluN2A (1.5). Per well of a 12-well plate, a mixture of plasmid DNA and calcium chloride (50 μl, 2.5 M) was added dropwise to 50 μl 2x HEPES buffered saline (E1200, Promega) and incubated for 20 min at room temperature in the dark. Coverslips containing neurons were transferred to a fresh 12-well plate containing 500 μl prewarmed culture medium supplemented with 1 mM kynurenic and followed by the addition of DNA/$CaCl_2$-HEPES solution. After a 90-minute incubation at 37 °C with 5% $CO_2$, the coverslips were transferred to a new 12-well plate containing culture medium, which had been pre-incubated at 37 °C and 10% $CO_2$. The 12-well plate was then incubated at 37 °C with 5% $CO_2$ for 20 min before being transferred back to the original 12-well plate (containing conditioned medium) and incubated at 37 °C with 5% $CO_2$ until the day of imaging.

**Electrophysiology**. The electrophysiology method utilised throughout was current measurements in whole-cell patch-clamp configuration. Hippocampal slices had a cut made between the CA3 and CA1 regions to reduce recurrent excitation during subsequent recordings of evoked synaptic transmission. Slices on their membrane were transferred to the chamber of an upright microscope (SliceScope Pro 2000, Scientifica), visualised using infrared light (780 nm) through an oblique condenser and 40 × 0.8 NA objective (LUMPLFLN40XW, Olympus) and captured using a mono charge-coupled device (CCD) monochrome camera (S-SciCam-00-EX, Scientifica) controlled using Scight software (Scientifica). The tissue was held down with a slice anchor and perfused at 2 ml/min with artificial cerebrospinal fluid (aCSF) solution containing (in mM): NaCl (125), KCl (2.5), $NaHCO_3$ (25), glucose (10), Na-pyruvate (1), $MgCl_2$ (1) and $CaCl_2$ (2). 5 μM 2-chloroadenosine (ab120037, Abcam) was added to prevent epileptic activity. Addition of other drugs depended on the experiment (see below). The aCSF was bubbled with carbogen (95% $O_2$, 5% $CO_2$) and maintained at 30 °C with an in-line heater (SM-4600, Scientifica) and heated chamber (HCS, ALA Scientific; TC-10, NPI Electronic). A bipolar tungsten stimulation electrode was positioned (LBM-7, Scientifica) in *stratum radiatum* to stimulate the Schaffer collateral pathway. 2–4 MΩ patch pipettes for whole-cell patch clamp recordings were filled with intracellular solution composed of (in mM): $CH_3SO_3H$ (120), CsOH (120), CsCl (20), $MgCl_2$ (2.5), HEPES (10), Na-ATP (4), Na-GTP (0.4), phosphocreatine disodium salt (5) and

EGTA (0.2) (pH 7.25). Osmolarity of intracellular and extracellular solutions was checked and adjusted on the day of each experiment and adjusted by addition of water or the major salt so that the aCSF osmolarity was 10–15 mOsmol/kg greater than the intracellular solution. To account for inter-slice variability in our recordings, two CA1 neurons were patched simultaneously—one transfected neuron identified by the Cre-GFP fluorescence (pE-300 white, CoolLED; 49018 EGFP long-pass, Chroma) and one neighbouring untransfected neuron. The intensity and polarity of the constant voltage stimulus (50 μs duration) was adjusted (IsoFlex, A.M.P.I.) to obtain reliable NMDA-EPSCs (in untransfected neurons) that were <250 pA (typically 10 V). The configuration of constant voltage stimulus enabled us to induce rapid changes in the potential difference across the electrodes with minimal influence of stray capacitance. Stimuli were delivered at an inter-sweep interval of 10 s. Signals were acquired with a MultiClamp 700B (Molecular Devices), low-pass filtered (4 kHz, 4-pole Bessel) and digitised (25 kHz) directly to hard disk using a USB-X Series Multifunctional DAQ interface (NI USB-6363, National Instruments). Hardware was controlled using Python-based ACQ4 software (v0.9.3)[77]. The series resistance was left uncompensated during data acquisition (but corrected offline, see below). Pairs of neurons were discarded during data collection if the difference in series resistance varied by more than ~8 MΩ.

For molecular replacement experiments (Figs. 2, 4 and S4), 10 evoked mixed AMPAR- and NMDA-EPSCs were recorded at a range of voltages (−100 to +20 mV in 20 mV increments). These holding potentials account for the liquid-liquid junction potential, which we calculated to be +10 mV for our solutions (PowerPatch Tools Igor Pro Tool collection). The aCSF included 50 μM picrotoxin, 10 μM gabazine as above, and 2 μM CGP52432 to inhibit GABA_B receptors. The intracellular solution was supplemented with 3 mM QX-314-Cl to attenuate currents through voltage-activated sodium and calcium channels. For each voltage, 10 evoked NMDA-EPSCs were recorded per neuron at an inter-sweep interval of 10 s. Transfections and recordings for GluN2A mutant molecular replacements were randomised and performed double blind. All molecular replacement experiments were carried out in slice cultures prepared from the left hemisphere since the measurements of NMDA-EPSC properties tended to show a lower variance compared to those in neurons of slices from the right hemisphere. The mutants selected for recordings in this study are known to have no more than 2.5-fold changes in magnesium potency[52,58]. We thus separated AMPAR- and NMDAR-components of evoked mixed EPSCs during the analysis using the following protocol: (1) The current traces at each holding potential were converted to conductance using Ohm's law; (2) The AMPAR-mediated conductance at −100 mV was subtracted from the conductance trace at +20 mV. The resulting conductance traces thus reflect AMPA-EPSCs (−100 mV) and NMDA-EPSCs (+20 mV).

For experiments where the NMDAR current was isolated pharmacologically (Fig. 5), the aCSF included 50 μM picrotoxin (ab120315, Abcam) and 10 μM gabazine (SR95531, Abcam) to block ionotropic gamma-amino-butyric acid receptors (GABA_ARs) and 10 μM NBQX (1044, Tocris Biosciences) to inhibit α-amino-3-hydroxy-5-methyl-4-isoxazolepropionic acid receptors (AMPARs). The membrane potential was raised from −75 mV to +20 mV at a rate of 0.5 mV per second to provide time for the Cs⁺ to block potassium (leak) conductances. For each neuron, 30 evoked NMDA-EPSCs were recorded.

Since the current–voltage relationship for AMPA-EPSCs_{-100 mV} and NMDA-EPSCs_{+20 mV} relative to the reversal potential (approximately 0) can be assumed linear, we performed offline corrections to compensate for series resistance[79]. The stimulus artefact was interpolated and the series resistance (Rs) and cell membrane capacitance (that were estimated from the test pulse in each recording) were used to perform offline series resistance compensation. For consistency, final, uncompensated current in each recording ($Rs_{final}$) was set to a constant 3 Mohm, where the % of compensation applied to each recording was calculated using the following equation.

$$Rs_{comp}(\%) = 100 \times \max\left(1 - \frac{Rs_{final}}{Rs}, 0\right)$$

Series resistance compensated recording traces were analysed using a custom python module (https://github.com/acp29/penn) within Stimfit software (v0.13 or 0.15.8)[80] or custom scripts in Matlab (available at https://github.com/acp29/Elmasri_GRIN2A). Decay of the EPSC currents was fit by a two-component exponential decay with offset initially using the Chebyshev algorithm. The weighted exponential time constant was calculated as the sum of component time constants multiplied by their fractional amplitude. Where the two-component fits failed, a single exponential was found to fit well.

**Imaging.** Transfected cultures of dissociated hippocampal neurons were imaged at DIV11. Neurons were imaged using an inverted microscope (AxioVert.A1, Zeiss), a 40 × 0.55 NA objective and a complementary metal-oxide-semiconductor (CMOS) camera (ProgRes GRYPHAX Arktur 8 MPix, Jenoptik). Images were acquired using GRYPHAX software (v2.2.0, Jenoptik). Coverslips were placed in ASCF composed of (in mM): NaCl (140), KCl (3), MgCl₂ (4), CaCl₂ (4), glucose (10), Na-pyruvate (1), HEPES (6), Na-HEPES (4) (pH 7.35, 300 mOsm). Neurons were imaged live for the GluN1-SEP signal at (LED λ_ex 470 nm) and the Homer1c-tdTomato signal at (LED λ_ex 530 nm). Five images (8-bit) were taken for each

coverslip and converted to grayscale in FIJI software (v1.53)[81,82]. Red and green images were aligned if necessary, with custom macros using a modified version of the Align_RGB_planes plugin available at https://github.com/acp29/pimage/tree/master/Penn/align_rgb). The background fluorescence was subtracted from both images, and puncta in the Homer1C-tdTomato signal were selected semi-automatically as regions of interest (ROIs). The ROIs were overlaid onto the GluN1-SEP signal and the mean pixel intensity was measured at these ROIs. Custom macros used for the measurement of spine fluorescence analysis are available at https://github.com/acp29/pimage/blob/master/Penn/pimage.ijm. The experimenter was blind to the experimental groups during image analysis, and the results were reproduced by multiple lab members.

**NEURON simulations.** Simulations were conducted in NEURON 7.5 software[83]. Simulations were carried out using a multicompartmental model of a CA1 pyramidal neuron developed by Graham and colleagues[69]. The model includes spatially distributed ion channels responsible for fast sodium (Na) currents, delayed rectifier potassium (K) currents, A-type potassium (K) currents, h-current, HVA R-type calcium (Ca²⁺) current, and Ca²⁺-activated medium-duration after-hyperpolarization (mAHP) potassium (K) currents. In all cases, simulations were conducted with a time step of 25 μs and at a temperature of 35 °C. Simulated synaptic inputs consisted of both AMPAR and NMDAR conductances described by the sum of two exponentials. AMPAR-receptor rise and decay time constants were set to 0.3 ms and 3 ms, respectively, and maximal conductance (g_max) was set to 500 pS. WT NMDAR rise and decay time constants were set to 3 ms and 80 ms respectively. The rise and decay time constants of NMDAR conductances were scaled by a factor of 1.5 or 2.0 to approximate the kinetics of GOF and LOF mutations on NMDA-EPSCs. The NMDAR gmax was set to 1 nS, 666 pS and 500 pS to emulate WT, GOF and LOF NMDARs respectively, with identical charge transfer. Activation of 40 or 200 excitatory synapses (as two-compartment spines) located randomly along the apical dendrites was used to simulate a barrage of synchronous synaptic events. For each experimental run, barrages of synaptic events were generated in trains of 5 stimuli at frequencies of 20, 40, 80 or 200 Hz. With the exception of the randomised distribution of activated synapses controlled by setting the random seed, the models and experiments that were conducted were all deterministic in nature.

**Statistics.** Data were collected until the number of cell pairs reached between a total of about 15–30 per mutation (or genotype). The experiment was repeated in 3–6 animals. The choice of which slices/animals to allocate to which mutation or genotype was random. The experimenter was blind to the mutation or genotype of the sample. Since it is possible that the occasional neuron we patched was not representative of the population (e.g. an interneuron, or had poor co-transfection with GRIN2A cDNA), we performed outlier detection prior to data analysis. Whole-cell properties and measurements of NMDA-EPSCs in matched transfected and untransfected where expressed as log_e response ratios and included in multi-variate outlier detection by robust principal component analysis (ROBPCA) using the LIBRA Matlab library[84,85] in Matlab 9.2 (R2017a, Mathworks). ROBPCA produces principal components (PCs) that are representative of the trend of the majority of the data with lower influence from extreme values. The number of PCs used was the number that explained 90% of the variance. Samples that were extreme orthogonal outliers, bad leverage points and were beyond the cut-off value of $\sqrt{X^2_{q,0.975}}$ were excluded from all subsequent analysis. Typically, about 2 recording pairs (range from 0 to 4) were excluded per group of data (e.g., per mutation). A summary of the sample sizes after multivariate outlier detection is presented in Table S4.

Statistical analysis was carried out in R (v 4.1.0) and RStudio (v 1.4). Each response datum from the patch-clamp recordings (in Figs. 2, 4, 5 and S4) was collected along with its assignment to (levels of) the following factors: mutation (WT, C436R, T531M, R518H, K669N, L812M) or genotype (wildtype, heterozygote, homozygote), transfection (unt. or tr., for untransfected or transfected respectively), and cell pair, slice, and animal (with unique identifiers for each to make the nested data hierarchy explicit). For the imaging experiments (in Fig. 3), each fluorescence measurement was collected along with its assignment to (levels of) the following factors: mutation (WT, C436R, T531M, R518H, K669N, L812M), protein (GFP-GluN1 or Homer1c-tdTomato), region of interest (ROI) and experimental repeat (with unique identifiers for each to make the nested and crossed data hierarchy explicit). Factors were appropriately assigned as fixed effects, or (nuisance) random effects. Response data from each experimental dataset was log_e-transformed and then fit by linear mixed effects regression with univariate linear mixed models (LMMs) using the lmer function from the R package lme4. Model fitting was set to optimise the restricted maximum likelihood (REML) criterion. The lmer formula for each random intercept model is reported in Figs. S1.1–S1.11 and follows the described convention for mixed models[86]. Where hypothesis testing was based on a priori clustering of the data (as in Fig. 1e), orthogonal contrasts (Table S1) were setup and applied before model fitting. Where hypothesis testing was based on testing the significance of trends across ordered (categorical) factors, polynomial contrasts were setup and applied before model fitting.

Model assumptions were evaluated by graphing plots of the residuals. Homoscedasticity was assessed by examining the scatter of the standardised model

residuals against the fitted values. Normality of the model residuals was assessed by (1) superimposing a histogram and kernel density estimate of the standardised model residuals with a standard normal distribution; and by (2) creating a quantile–quantile (Q–Q) plot of sample quantiles against normal quantiles together with 95% tail-sensitive confidence bands. Model outliers were assessed from the above plots of the standardised model residuals and the degree to which they were influential was assessed using a stem-and-leaf plot of the Cook's distances calculated using the `hlm_influence` function from the `HLMdiag` package. Cook's distances were all <0.15, far below the threshold for being considered influential (1.0). Graphs were created using functions from the `ggplot2` package.

The LMM results for the fixed effects were summarised as ANOVA tables using Type III Wald $F$-tests and with degrees of freedom calculated by the Kenward-Roger method using the Anova function from the car package. To summarise the random effects structure, the variance of each group level was used to calculate its respective intraclass correlation coefficient (ICC) using the `icc` function from the `performance` package. When *posthoc* tests were requested, they were pairwise comparisons with type I error rate controlled by the Westfall stepwise procedure and were calculated using the `multcomp` R package. Parameter estimates and effect sizes are reported in the main text and figures as response ratios (expressed as %) with 95% confidence intervals calculated using functions from the `emmeans` package. Effect sizes were standardised by conversion to correlation coefficients using the `t_to_r` function from the `effectsize` package and reported in the figure legends. Details of LMM results and diagnostics are provided in the supplement (Fig S1). All summaries of statistics annotated in graphs have used the following convention: $ns$ = not significant (at $\alpha = 0.05$), $*$ = $p < 0.05$, $**$ = $p < 0.01$, $***$ = $p < 0.001$. All reported $p$-values are from two-tailed tests. Code and data relating to the analysis are provided at https://github.com/acp29/Elmasri_GRIN2A.

To evaluate the strength of evidence for the null hypothesis for omnibus ANOVA tests of the fixed effects, Bayes Factors ($BF_{10}$) were calculated using the `anovaBF` and `bayesfactor_inclusion` functions from the `BayesFactor` (v0.9.12-4.3) and `bayestestR` packages. $BF_{10}$ indicates the likelihood ratio of evidence for ($BF_{10} < 1$), or against ($BF_{10} > 1$), matched (null) models. Default (Cauchy) priors for fixed and random effects had the usual scales of 0.5 ('medium') and 1 ('nuisance') respectively.

Confidence intervals and $p$-values for Kendall's tau-b ($\tau_b$) (non-parametric) correlation coefficients were calculated in Matlab using *iboot*[87], now part of the Octave statistics-bootstrap package (https://gnu-octave.github.io/packages/statistics-bootstrap). Central coverage of bias-corrected and accelerated (BCa) intervals was calibrated to reduce small sample bias using a double bootstrap procedure, with 20,000 and 200 resamples for the first and second bootstrap respectively.

**Structural modelling.** Structural modelling was performed on GluN2A in a pseudo full-length human model of a triheteromeric NMDA receptor. The coordinates of the triheteromer structure of *Xenopus laevis* GluN1/2A/2B (4.5 Å, 5uow) were downloaded from the Orientations of Proteins in Membranes (OPM) database[88]. GalaxyFill was used to fill in missing main chain and sidechain atoms and loops of each protomer[89]. The protomers then reassembled and steric clashes removed by brief energy minimisation using steepest descent. This structure had a root mean squared deviation of CA atoms (CA-RMSD) of 0.08 Å (from 5uow) and was used as scaffold for modelling the ligand-bound human triheteromeric receptor in Modeller (v 9.19)[90] using multiple high-resolution templates: 4nf8, 5fxh, 5fxi, 5i57, 5tpw, 5tpz and 5un1. The *automodel* function was used to create 400 models using 1000 iterations of the variable target function method (VTFM, *autosched.slow*) followed by Simulated Annealing Molecular Dynamics (SA-MD) refinement (*refine.very_slow*). Each model was assessed using three statistically optimised atomic potentials (SOAP, soap_protein_od, soap_pp and soap_peptide) to assess sidechain packing, subunit interaction surfaces and amino acid ligands[91]. Following transformation of each potential to Z-scores, models were accepted whose scores were more negative than -1 standard deviation. Of the accepted models, the one with the lowest combined potential was taken forward for refinement by restrained molecular dynamics in CHARMM[92] using the CHARMM36m forcefield[93]. In brief, alternative protonation states for Glu, Asp and Lsn (at neutral pH) were identified using PROPKA3.1[94] and patched along with disulphide bridges and C-terminal $N$-methylamide. Hydrogens were added using H-BUILD and the structure was embedded in a heterogeneous dielectric generalised Born (HDGB) implicit membrane (v3)[95] with infinite cut-offs for non-bonded interactions. Solvent accessible residues within the channel were excluded from the low dielectric using the EXCLGB parameter[96]. Within the implicit solvent and membrane environment, the structure was subjected to a series of restrained energy minimisation and molecular dynamics using a protocol closely resembling the Local Protein Structure Refinement via Molecular Dynamics Simulations (locPREFMD)[97]. Molecular dynamics simulations were performed using the velocity Verlet integrator with a 1 fs time step and SHAKE constraints applied on bonds involving hydrogen.

*In silico* point mutations were generated in the NMDA receptor model using Modeller 9.19 by a combination of conjugate gradient minimisation and molecular dynamics simulated annealing[98] (https://salilab.org/modeller/wiki/Mutate model). The procedure was repeated with different random seeds to generate 100 different unique models and the model with the lowest energy was selected.

All modelling and refinement steps were accomplished on a Dell Precision T7910 workstation with dual Intel Xeon processors (E5-2687W @ 3.10 GHz, 10 cores each), Structures were visualised (and figures created) in Pymol (The PyMOL Molecular Graphics System, Version 1.7.7.6 Schrödinger, LLC.). Scripts and data relating to the structural modelling are provided at https://github.com/acp29/Elmasri_GRIN2A.

**Reporting summary**. Further information on research design is available in the Nature Research Reporting Summary linked to this article.

## Data availability
Source data used for statistical analysis and graphs are deposited at https://github.com/acp29/Elmasri_GRIN2A. Source data for graphs are also available in Supplementary Data 1–6.

## Code availability
Code used for analysis and modelling is deposited at the following repositories: https://github.com/acp29/penn, https://github.com/acp29/Elmasri_GRIN2A, and https://github.com/acp29/pimage/tree/master/Penn.

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

## Acknowledgements

The authors gratefully acknowledge the following kind gifts: pCMV Cre-GFP from Christophe Mulle (IINs, Bordeaux, France), pCMV GluN2A from Kasper Hansen, Hongjie Yuan, and Stephen Traynelis (Emory, Atlanta, Georgia, USA) and pCaMKII-αHomer1c-tdTomato from Daniel Choquet (IINs, Bordeaux, France). The PhD studentship of M.E. was funded on a University of Sussex Life Sciences School studentship. A.C.P. was funded on MRC Career Development Award (MR/M020746/1), in which W.A. and M.E. were funded as post-docs.

## Author contributions

M.E. performed cloning and electrophysiology experiments and analysis related to Figs. 2, 4 and 5, and S3, 4. D.M.V.D, D.W.H. and E.E.B. performed imaging experiments and analysis related to Fig. 3 and S2. G.W. and E.E.B. performed simulations related to Fig. 6. W.A. performed some electrophysiology experiments related to Fig. 5. E.K. performed cloning experiments related to Figs. 2, 3 and 4. K.S. provided all genetically-altered mice. A.C.P. obtained funding, designed the experiments, analysed data related to all figures, performed molecular modelling and analysis related to Fig. 1 and S5, performed all statistical analysis and wrote the paper. Design of experiments was discussed with all authors.

## Competing interests

The authors declare no competing interests.
