## [Peer Review File · Communications Biology]

Reviewers' comments:

Reviewer #1 (Remarks to the Author):

COMMSBIO-20-2559-T

This is a complex manuscript investigating synaptic properties afforded by genetic modifications in the GluN2A subunit. The authors used whole cell recording of stimulus-evoked NMDA currents at different holding potentials in mouse organotypic hippocampal slice cultures under visual control and applied complex statistical evaluation including PCA. The study excellently complements findings of GRIN2A point mutations in various human epilepsies

There are few very minor concerns:

Introduction:

(1) For completeness the authors should at least mention and reference GluN3/GRIN3

Methods

(2) Rate of perfusion with ACSF needs to be provided.

(3) Why was constant voltage used for stimulation? Please justify.

Reviewer #2 (Remarks to the Author):

Recent years have seen a booming in the identification of clinically-relevant mutations associated with brain disorders. In particular, *grin2a*, the gene encoding the NMDA receptor GluN2A subunit, has emerged as a major risk factor for a variety of neurodevelopmental disorders, with epilepsy constituting a major core phenotype. There is currently great interest, both from a basic research and therapeutic point of view, to understand how the mutations affect receptor and circuit function, and ultimately behavior and cognition.

In this work, using a molecular replacement strategy on organotypic mouse hippocampal slices, the authors study the impact on synaptic function of five *Grin2A* missense mutations previously identified in the clinic. All five mutations cluster in the gating core region of the receptor (LBD+TMD) and appear to act as gain- or -loss-of-function mutations, at least according to studies on recombinant receptors expressed in heterologous systems. Despite these diverging effects, the authors show that all five mutations result in a surprisingly similar synaptic phenotype, combining decreased peak current with prolonged decay kinetics. The resulting net effect is a lack of effect on charge transfer as assessed by integrating NMDA-EPSC currents.

This is a well performed study involving state-of-the-art manipulations of synaptic receptor function (molecular replacement using floxed *Grin2a* and *Grin 2b* mouse lines, dual patch-clamp recording allowing for internal control experiments). Overall, the authors have accumulated a large data set that provides novel insights into the functional impact of clinically-relevant NMDAR mutations at the synapse level. That said, the conclusions of this work may not be as broad in scope as the authors suggest, and a number of limitations of the approach regarding our understanding of disease mechanisms should be better emphasized. Moreover, a few key additional experiments could help moving beyond the apparent lack of functional effects in basal synaptic transmission and provide a cellular and network basis for behavioral defects.

Main points:

- Generality of the finding. The authors draw rather strong conclusions on common features and impacts on synaptic function of disease-linked GluN2A-NMDAR mutations. Hundreds of GluN2A mutations in the clinic have been identified yet the authors based their conclusions on just five mutations. Out of these five, three basically express very poorly or are non-functional (R518H)

and show strong GluN2B compensation, and two are gain-of-function (with established slower glutamate deactivation time course). It is quite conceivable that other GluN2A mutations, in particular loss-of-function mutations that retain expression and functionality, behave quite differently (with no or little GluN2B upregulation). Similarly, other gain-of-function mutations that do not alter deactivation kinetics but, for instance, increase P_o or glycine sensitivity, may yield different phenotypes. The authors should better take into account this potential heterogeneity to avoid overgeneralizing the significance of the present results.

- GluN2B compensation. Interestingly, the data strongly point to GluN2B subunits compensating for mutant 'dead' GluN2A subunits (subunit that show little or no expression or lack functionality). Because GluN2B and GluN2A subunits couple to distinct downstream signaling pathways, this replacement is expected to drastically impact neuronal signaling such as long-term synaptic plasticity. It would be most interesting and informative to assess the impact of the GluN2A mutations on NMDAR-dependent LTP and LTD, two forms of long-term synaptic plasticities that are well-characterized at hippocampal CA1 synapses and widely thought as major cellular substrates of memory storage and cognitive performance.

- Complexity of the genotype-phenotype relationship. As a net effect on synaptic charge transfer, the mutations studied here appear silent displaying no or minimal impact (in comparison to WT). This is highly intriguing and raised fundamental questions about the underlying disease mechanisms caused by the mutations *in vivo*. Because GluN2A-NMDARs are expressed in diverse neuronal types (both excitatory and inhibitory) in a developmental and region specific manner, and participate in distinct NMDAR populations (including extrasynaptic and presynaptic), the authors should better acknowledge and comment on the possibility that GluN2A mutations may alter circuit function through a diversity of mechanisms (most likely leading to altered E-I balance). The last computational figure provides hints on how NMDA-EPSCs with similar charge transfer capacity but distinct time course may lead to different summation properties. This is an interesting prediction that can be experimentally tested. Providing this experimental validation would help resolving the conundrum of disease-causing mutations being functionally 'silent' and help rationalize how cellular and network level alterations may cause behavioral and cognitive deficits (see also previous point for effects depending on subunit molecular identity rather than charge transfer only).

Other points:

- The authors should better explain the rationale underlying their choice of mutations (5 among 9 listed in groups 1 3 and 4; Fig 2E).
- To avoid any confusion, the authors should further emphasize that patients have a single copy of the mutated GluN2A subunits while in their experiments both alleles are mutated maximizing potential effects.
- The GluN2A/GluN2B is not the only critical determinant of NMDA-EPSC time course. Splicing of the GluN1 subunit, i.e. relative expression of the GluN1-1b vs GluN1-1a splice variants, is another important contributing factor (Liu et al., PNAS, 2019). One cannot exclude that altered GluN2A expression or functionality affect GluN1 splicing efficiency as a downstream effect. This should be mentioned.
- Triheteromeric receptors: It would be interesting to test the effect of expressing GluN2B and GluN2A-L812M in the same *Grin2a*^{-/-b} animal. Effects of expressing GluN2B, GluN2A-L812M or both in the same cellular context could then be directly compared.
- Figure 3: The statistical significance strongly differs between panels D & E (2 or 3 stars in panel D vs no or 1 star in panel D). Does this mean that differences between WT and untransfected neurons are large enough to confound biological significance?
- p5, 1st paragraph: the way the authors report changes in peak amplitudes, here and elsewhere in the manuscript, is misleading. Indeed, a reduction by a factor of about 0.6, as stated, suggest a potentiation effect. Change in peak current amplitudes should be expressed as changes in peak ratio of % of current change (increase or decrease).
- p3, 1st paragraph: the reference to Figure 1B is incorrect.
- A reference to Figure 4D should be added page 9.
- Page 3, 'can be unpredictable and no one anti-epileptic drug (AED)', the word 'one' should be deleted.
- Page 3, "Indeed, an untested hypothesis is that GluN2B could compensate for loss-of-function mutants of GluN2A (ref 40)": ref 40 says that there is no sign of compensation by GluN2B.

- Page 4, 'Figure 1A' should be 'Figure 2A'.
- Page 8, 'Table II' should be 'Table III'.
- Page 9, "Charge transfer of mutant NMDAR-EPSCs based on the Swanger and others": the model of Swanger predicts theoretical EPSCs mediated by GluN2A subunits only. One can reasonably expect a different outcome in real neurons expressing a mixture of receptor populations.
- Page 13, 'that there is not complete compensation by GluN2B', 'no' instead of 'not'.
- Page 14, 'Figure 2' end of 2nd paragraph should be 'Figure 1'.
- Methods: Is there a reason for why series resistances were not compensated for?
- Methods: Page 22 please clarify the mathematical expressions ' $\log(\text{ratio}) \sim 0 + \text{genotype} + (1|\text{animal/slice})$ '. Also expression on top of page 23.
- Supplementary table II: It is unclear to what "n,slices,animals" refers to, especially when compared to the numbers indicated on the violin plots (in all figures).

Response to reviewers: COMMSBIO-20-2559-T

We thank the reviewers for their constructive and insightful comments. We begin by summarizing the steps we have taken to address the concerns of the reviewers and then provide a point-by-point response to each of the reviewers. To help reviewers navigate through the changes in the manuscript, we have added line as well as page numbers to the article and we refer to line numbers in our responses. In addition, we provide a copy of the article with changes marked up.

In response to the reviewer's comments, we have focussed on performing experiments, analysis and changes to the manuscript that enhance the rigor of the study and clarified the study's scope. Here is a summary of steps taken in response to reviewer comments:

- We have performed new computational experiments that supersede the computational work presented in the original manuscript. The CA1 model in our new NEURON simulation includes many more active conductances and models intracellular Ca^{2+} dynamics (including Ca^{2+} through NMDA receptors and voltage gated Ca^{2+} channels). We think that the new data provides: 1) a more compelling argument that mutant-like changes in NMDA receptor kinetics can still affect synaptic excitability even if charge transfer through the NMDA receptor is indifferent; and 2) deeper insight into the potential outcome of mutant-like changes in NMDA receptor properties (including differences in neuronal spiking, the accumulation of spine Ca^{2+} and plateau potentials). The data is presented in **Figure 6** of the updated manuscript.
- In light of comments about our manuscript being complex we have condensed and restructured manuscript results section to simplify the narrative.
- Online series resistance compensation was not used when the data was acquired for reasons that we detail in our point-by-point responses below. To address the reviewer comment, we have reanalysed all the electrophysiology data so that the analysis of the data now included offline series resistance compensation. The new manuscript now has updated data sets relating to the following figures and tables: **Figures 2, 4, 5, S1 (except (S1.3)), S4, and Table S4**
- Many of the comments from reviewer 2 requested extending the scope of the study. We believe that this is, in part, because our discussion didn't include or emphasis some caveats of our study and that our conclusions were at times overly broad. We have revised some of the content, added some discussion and changed the wording of our conclusions in the abstract and discussion accordingly.
- With the new data sets we took the opportunity to revise the data presentation and statistical analysis (with the Bayesian statistics now complemented with more conventional Frequentist statistics). Below is Figure S4, an example of the method of data presentation. To make it clearer how data was handled and what the plotted data represents, we now plot the simultaneously recorded electrophysiology measurements from untransfected (unt.) and transfected (tr.) neurons, and indicate their matching/pairing by connecting the data points with a line (**ai** in the example below). Underneath the data points, we also show crossbars denoting the mean and 95% confidence intervals (from the fitted linear mixed model). While we sometimes plotted this information in the original manuscript but as scatter plots for transfected (y) vs untransfected (x), these new graphs are more space efficient. In an accompanying graph (**aii**) we plot the response ratio (tr./unt. ratio represented as a %, e.g. tr./unt. 0.56 is 56%) and annotate the graph with results of post hoc tests or orthogonal contrasts from the fitted model. A summary of the output and assumption testing of the linear mixed models is provided in **Figures S1.1 – 1.11**, along with a breakdown of samples sizes in **Table S4** and information pertaining to orthogonal contrasts in **Table S1**. In the text and figure legends we report the response ratios and standardized effect sizes respectively (for the latter we chose the more familiar correlation coefficient as a standardized effect size).

Figure S4: GluN2A mutations are not associated with large effects on AMPAR-EPSCs.

NMDA-EPSC_{+20 mV} peak amplitudes in *grin2a*^{fl/fl} (untransfected) neurons and *grin2a*^{-/-} neurons rescued with human GluN2A WT, LOF (C436R, T531M or R518H) or GOF (K669N or L812M) mutants (transfected). **ai**) Data points of measurements made in individual neurons. Matched data points, for simultaneously recorded untransfected and transfected neurons, are connected by a line. **aii**) Response ratios (transfected/untransfected) are expressed as a percentage and plotted for each pair of transfected-untransfected neurons. Crossbars in **i**) and **ii**) show the estimated marginal means with 95% confidence intervals backtransformed from the linear mixed models (Figure S11). Hypothesis tests are orthogonal contrasts based on *a priori* clustering of the mutations in Figure 1e. Standardized effect sizes (*r*) for comparisons of each mutant with WT for response ratios of peak amplitude in **aii** were .09, .07, .05, -.08 and -.08 (*N* = 125).

Response to reviewers: COMMSBIO-20-2559-T

Reviewers' comments:

Reviewer #1 (Remarks to the Author):

COMMSBIO-20-2559-T

This is a complex manuscript investigating synaptic properties afforded by genetic modifications in the GluN2A subunit. The authors used whole cell recording of stimulus-evoked NMDA currents at different holding potentials in mouse organotypic hippocampal slice cultures under visual control and applied complex statistical evaluation including PCA. The study excellently complements findings of GRIN2A point mutations in various human epilepsies

We thank the reviewer for their comment that “[this] *study excellently complements findings of GRIN2A point mutations in various human epilepsies*”. With respect to their remark about the manuscript being ‘complex’, we have gone to a great deal of effort in revising the structure and presentation in the manuscript to make the message of the paper clearer and make it more transparent how data was analysed.

There are few very minor concerns:

Introduction:

(1) For completeness the authors should at least mention and reference GluN3/GRIN3
We have now mentioned and included GluN3 in the introduction at page 2 lines 12-13.

Methods

(2) Rate of perfusion with ACSF needs to be provided.

We have added the rate of aCSF perfusion to the methods section at page 20 line 13. The rate of perfusion was 2 ml/min.

(3) Why was constant voltage used for stimulation? Please justify.

We assume that the reviewer is referring to the stimulation of presynaptic fibres for evoking synaptic transmission, as opposed to application of electrical stimulation for single-cell electroporation. In our experience, for tungsten/platinum wire electrodes, as we used here, the choice between constant current and constant voltage isn't so critical so long as significant bubbles do not form on the electrodes (which would affect the resistance between the electrodes). We didn't experience this problem at the relatively low stimulation voltages (~10 V) used in our relatively short experiments. We find that briefer, sharper stimulus duration can be achieved with constant voltage because it will deliver as much current as is needed to reach the desired square pulse in the potential difference between the electrodes – with the excess current supercharging stray capacitance. With constant current, the fixed amount of current will first need to charge stray capacitance in the system, meaning that the voltage (and current passing) between the electrodes will be filtered, thus requiring longer stimulus duration. We have added a comment in the methods section to this effect at page 20 line 31 to page 21 line 2.

There are situations where we find that constant current can be more appropriate. If one is using a monopolar glass electrode stimulator, constant current is the better choice since it provides more stable stimulation in the face of changes in series resistance. For very long recordings (e.g. > 1 hour) with frequent stimulation using monopolar patch stimulation electrodes, as a rule our lab would use constant current instead. This was not the situation in the experiments described herein.

Reviewer #2 (Remarks to the Author):

Response to reviewers: COMMSBIO-20-2559-T

Recent years have seen a booming in the identification of clinically-relevant mutations associated with brain disorders. In particular, *Grin2a*, the gene encoding the NMDA receptor GluN2A subunit, has emerged as a major risk factor for a variety of neurodevelopmental disorders, with epilepsy constituting a major core phenotype. There is currently great interest, both from a basic research and therapeutic point of view, to understand how the mutations affect receptor and circuit function, and ultimately behavior and cognition.

In this work, using a molecular replacement strategy on organotypic mouse hippocampal slices, the authors study the impact on synaptic function of five *Grin2a* missense mutations previously identified in the clinic. All five mutations cluster in the gating core region of the receptor (LBD+TMD) and appear to act as gain- or -loss-of-function mutations, at least according to studies on recombinant receptors expressed in heterologous systems. Despite these diverging effects, the authors show that all five mutations result in a surprisingly similar synaptic phenotype, combining decreased peak current with prolonged decay kinetics. The resulting net effect is a lack of effect on charge transfer as assessed by integrating NMDA-EPSC currents.

This is a well performed study involving state-of-the-art manipulations of synaptic receptor function (molecular replacement using floxed *Grin2a* and *Grin2b* mouse lines, dual patch-clamp recording allowing for internal control experiments). Overall, the authors have accumulated a large data set that provides novel insights into the functional impact of clinically-relevant NMDAR mutations at the synapse level. That said, the conclusions of this work may not be as broad in scope as the authors suggest, and a number of limitations of the approach regarding our understanding of disease mechanisms should be better emphasized. Moreover, a few key additional experiments could help moving beyond the apparent lack of functional effects in basal synaptic transmission and provide a cellular and network basis for behavioral defects.

We thank the review for highlighting that *“there is currently great interest... [in understanding] how [GRIN2A] mutations affect receptor and circuit function”*. Indeed, we agree that this is an important and very topical research area. We also thank the reviewer for commenting that our work represents a *“well performed study involving state-of-the-art manipulations of synaptic receptor function”*. We also agree with the reviewer that our extensive data set provides *“novel insights into the functional impact of clinically-relevant NMDAR mutations at the synapse level”*.

The reviewer has pointed out that the breadth in scope of our conclusions are a bit exaggerated and we have addressed this concern (detailed below). We emphasize that the providing a broader cellular and network basis for behavioural defects is beyond the scope of this study and we have adapted wording in the discussion to clarify to readers what inferences can be made directly from our results. We have revised the final sentence of the abstract (**page 1 lines 24-26**) in recognition that there is more direct and immediate impact to our research findings; it now reads: *‘An implication of this work is that investigations beyond establishing the molecular defects of GluN2A mutations are much needed to understand their impact on synaptic excitability.’*

We would like to emphasize though that in the title: *“Common synaptic phenotypes arising from diverse mutations in GRIN2A”*, we are not saying that the reported synaptic phenotypes arise from all types of mutations, we are simply saying that there are similar synaptic phenotypes *arising* from a set of mutations, which have diverse molecular defects.

Main points:

- Generality of the finding. The authors draw rather strong conclusions on common features and impacts on synaptic function of disease-linked GluN2A-NMDAR mutations. Hundreds of GluN2A mutations in the clinic have been identified yet the authors based their conclusions on just five mutations. Out of these five, three basically express very poorly or are non-functional (R518H) and show strong GluN2B compensation, and two are gain-of-function (with established slower glutamate deactivation time course). It is quite conceivable that other GluN2A mutations, in particular loss-of-function mutations that retain expression and functionality, behave quite differently (with no or little GluN2B upregulation). Similarly, other gain-of-function mutations that do not alter deactivation kinetics but, for instance, increase P_o or glycine sensitivity, may yield different phenotypes. The authors should better take into account this potential

Response to reviewers: COMMSBIO-20-2559-T

heterogeneity to avoid overgeneralizing the significance of the present results.

We have added some discussion on the point that the review kindly makes, at page 13 line 27 to page 14 line 3.

- GluN2B compensation. Interestingly, the data strongly point to GluN2B subunits compensating for mutant 'dead' GluN2A subunits (subunit that show little or no expression or lack functionality). Because GluN2B and GluN2A subunits couple to distinct downstream signaling pathways, this replacement is expected to drastically impact neuronal signaling such as long-term synaptic plasticity. It would be most interesting and informative to assess the impact of the GluN2A mutations on NMDAR-dependent LTP and LTD, two forms of long-term synaptic plasticities that are well-characterized at hippocampal CA1 synapses and widely thought as major cellular substrates of memory storage and cognitive performance.

The reviewer makes a valid point here and it is indeed an area worthy of discussion and future research. Native GluN2B compensating for mutant 'dead' GluN2A subunits will lead to the relative abundance of GluN2A and B protein at the synapse to change. One can reasonably expect differences then in the ability for synaptic NMDA receptors to activate signalling pathways uniquely coupled to the C-terminal domains of GluN2A or B subunits. Indeed, if this is the case then the "synaptic phenotype" for GluN2A mutants may be more different between LOF and GOF mutations than appears from our examination of the NMDA and AMPA components of basal excitatory synaptic transmission alone. We now freely acknowledge this in the discussion, at page 14 lines 14-20.

The study as it stands represents a huge amount of work mostly by a single PhD student (and subsequently postdoc) who has now left the lab, having (manually) obtained patch-clamp recordings from 910 neurons (half of these being transfected) for this study (see Table S4). Investigating LTP and LTD would be a sizeable effort that unreasonably extends the scope of the already substantial study presented in the manuscript. We are delighted though with the reviewer's enthusiasm for the plasticity experiments to follow up on the GluN2B compensation as it represents an encouraging sign that our work has raised progressive questions for further investigation in future studies.

- Complexity of the genotype-phenotype relationship. As a net effect on synaptic charge transfer, the mutations studied here appear silent displaying no or minimal impact (in comparison to WT). This is highly intriguing and raised fundamental questions about the underlying disease mechanisms caused by the mutations in vivo. Because GluN2A-NMDARs are expressed in diverse neuronal types (both excitatory and inhibitory) in a developmental and region specific manner, and participate in distinct NMDAR populations (including extrasynaptic and presynaptic), the authors should better acknowledge and comment on the possibility that GluN2A mutations may alter circuit function through a diversity of mechanisms (most likely leading to altered E-I balance). The last computational figure provides hints on how NMDA-EPSCs with similar charge transfer capacity but distinct time course may lead to different summation properties. This is an interesting prediction that can be experimentally tested.

Providing this experimental validation would help resolving the conundrum of disease-causing mutations being functionally 'silent' and help rationalize how cellular and network level alterations may cause behavioral and cognitive deficits (see also previous point for effects depending on subunit molecular identity rather than charge transfer only).

The reviewer raises an excellent point here and while role of mutant NMDA receptors in different neuronal types and subsynaptic locations has been something we've thought about a lot, it is not something that we discussed in the original manuscript and to address this with rigor would be best achieved with reporter mouse lines that enable the experimenter to target specific interneuron types (e.g. Parvalbumin interneurons). We have rightfully included a paragraph of discussion on the topic and include a citation to some excellent relevant, recent work by D. Wyllie's lab that have gone some way to addressing this question. The new discussion content can be found at page 15 lines 19-24.

With respect to the effect of mutant-like NMDA-EPSCs on EPSP summation properties, we also recognise that the simulations in the final figure of the original manuscript were limited in scope and were performed with a very simple model of a CA1 neuron (with glutamate receptor and I_h conductances only and without explicitly modelling spines). We have now adopted use of a computational model of CA1 neuron that includes a much wider range of active conductances, dendritic spines, intracellular calcium

Response to reviewers: COMMSBIO-20-2559-T

(with calcium buffering) and NMDA receptor calcium permeability. Synapses were randomly positioned over the areas of dendrite typically innervated by the Schaffer collaterals and simulations were repeated using 10 different random seeds. The key aim of this set of simulations was to establish whether differences in neuronal excitability can arise from (mutant-like) NMDA-EPSCs, which differ in kinetics but not charge transfer. This logically follows our experimental finding. The original computational results have been replaced with the new simulation results and are presented in **Figure 6** and **page 11 line 16 to page 12 line 30** of the revised manuscript.

A more extensive evaluation to ascertain in what ways exactly the NMDA receptor mutations, as a whole, affect neuronal and circuit function is beyond the scope of this study. In the words of the reviewer, we have already “*accumulated a large data set*” that in itself “*provides novel insights into the functional impact of clinically-relevant NMDAR mutations at the synapse level*”, which indeed reflects the intended scope of our study.

Other points:

- The authors should better explain the rationale underlying their choice of mutations (5 among 9 listed in groups 1 3 and 4; Fig 2E).

We thank the reviewer for voicing this thought, which I’m sure other readers will have. It is not the case that we have performed experiments on all nine and then only reported data from 5 of them. 9 mutations would be too many to study in this piece of work so we had to make a choice on which ones to pursue. We cloned representative mutations from the different clusters that had most extensive functional characterisation published at the time the project was started. During acquisition of data for these mutants, more reports of data for LOF and GOF mutants were published which we included in the analysis summarised in Figure 1. We have added a sentence to this effect in the relevant results section at **page 5 lines 25-28**.

- To avoid any confusion, the authors should further emphasize that patients have a single copy of the mutated GluN2A subunits while in their experiments both alleles are mutated maximizing potential effects.

We thank the reviewer for raising this important point. I hope that the reviewer appreciates our ethical grounds and rationale for performing our rescue experiments in the homozygous context. A lot of the effects we see in homozygous knockout *grin2a* neurons are also observed in heterozygous knockout neurons, albeit with smaller effect size (**Figure 5** of the updated manuscript). To increase the statistical power of our experiments without increasing the number of recordings (and thus animals), we decided to perform the mutant rescue experiments in homozygous knockout neurons. We expect that the effects in heterozygotes to be a bit smaller, like we observed for the heterozygote knockout allele (**Figure 5**), however we do appreciate that unexpected effects may arise for certain mutations in the context of expression alongside a wildtype allele. We have extended our discussion on this in the updated manuscript, at **page 15 lines 9-10**.

- The GluN2A/GluN2B is not the only critical determinant of NMDA-EPSC time course. Splicing of the GluN1 subunit, i.e. relative expression of the GluN1-1b vs GluN1-1a splice variants, is another important contributing factor (Liu et al., PNAS, 2019). One cannot exclude that altered GluN2A expression or functionality affect GluN1 splicing efficiency as a downstream effect. This should be mentioned.

We thank the reviewer for this suggestion and indeed we cannot exclude this possibility. We would like to highlight that the manipulations of NMDA receptors are in no more than 10 neurons per hippocampal slice (at the very most) so any downstream effects of GluN2A would need to be cell autonomous. While we consider that cell autonomous regulation of GluN1 splicing by GluN2A is unlikely we have now mentioned the possibility of this in the discussion at **page 14 lines 20-23**.

- Figure 3: The statistical significance strongly differs between panels D & E (2 or 3 stars in panel D vs no or 1 star in panel E). Does this mean that differences between WT and untransfected neurons are large enough to confound biological significance?

We thank the reviewer for raising this point. This is an issue where the reader interprets comparisons between two effects without directly comparing them. This issue is discussed well here at

Response to reviewers: COMMSBIO-20-2559-T

<https://elifesciences.org/articles/48175>. The apparent discrepancy likely arises because of the greater variance associated the propagation of error when making the comparison of untransfected/transfected ratios between mutant and WT. An equivalent way to look at this is to test the statistical interaction in a two-way (mixed) ANOVA on the \log_e -transformed response (in particular example raised by the reviewer, the response is the peak amplitude). The interaction, mutation x transfection, then tests whether the effect of transfecting *GRIN2A* on NMDA-EPSC response depends on the genotype. Studying interaction effects is renowned to have lower power than studying main effects, but we do gain some power from having the untransfected cell as an internal control (as illustrated below). In order to limit the amount of unexplained variance when evaluating statistical interactions (and thereby increase the reliability and power of our statistics) we routinely: 1) use a pooled variance across the levels of genotype to obtain less noisy estimates of the residual variance (as is standard in ANOVA), and 2) account for nuisance variance between cell pairs, slices and animals by incorporating them as random intercepts in mixed model. We have now reanalysed the data in figures 2, 4 and 5 (in the new manuscript), in a way that more explicitly illustrates the point **that what we are interested in the interaction, where comparisons are with wildtype but take into account the internal control**. Because setting the priors by the Bayesian multilevel modelling approach was becoming more tricky now with fitting to 2 fixed effects and an interaction, we have taken a (more familiar) frequentist approach by fitting linear mixed models. For hypothesis testing using Bayes factors, we used the default diffuse priors from the BayesFactor R-package. We have revised the presentation of the data and deposited the R markdown and knitted HTML output on an open access server (https://github.com/acp29/Elmasri_GRIN2A) for the sake of openness and transparency.

- Triheteromeric receptors: It would be interesting to test the effect of expressing GluN2B and GluN2A-L812M in the same *Grin2a*^{-/-} animal. Effects of expressing GluN2B, GluN2A-L812M or both in the same cellular context could then be directly compared.

If we understand correctly, the experiment proposed by the reviewer is to test the hypothesis we raised in part of the discussion with respect to the effect that triheteromerization of GluN2B and GluN2A-L812M. We too would find this experiment 'interesting' and *it is now an avenue we are pursuing in another study*. In addressing other reviewer comments about the manuscript/study complexity we have refocused the discussion of the revised manuscript and, in doing so, no longer discuss triheteromerization in order to make room for more pertinent areas of discussion. We hope that the reviewer respects our response in the context of the ways we have tried to address their other comments and with having a manuscript of reasonable length.

- p5, 1st paragraph: the way the authors report changes in peak amplitudes, here and elsewhere in the manuscript, is misleading. Indeed, a reduction by a factor of about 0.6, as stated, suggest a potentiation effect. Change in peak current amplitudes should be expressed as changes in peak ratio of % of current change (increase of decrease).

We thank the reviewer for highlighting this area of ambiguity in our reporting. We now simply report the % of mutant relative to wildtype. For example, "*the NMDA-EPSC peak amplitudes for LOF mutants relative to WT (100%) were 35% for C436R, 38% for T531M and 37% R518H*". We hope that this is easier for the reader to follow.

- p3, 1st paragraph: the reference to Figure 1B is incorrect.

The figures have been reorganised and the in-text references to figures corrected in the revised manuscript.

- A reference to Figure 4D should be added page 9.

The figures have been reorganised and the in-text references to figures corrected in the revised manuscript.

- Page 3, 'can be unpredictable and no one anti-epileptic drug (AED)', the word 'one' should be deleted. Thank you for spotting this, we have deleted 'one' from this sentence.

Response to reviewers: COMMSBIO-20-2559-T

- Page 3, “Indeed, an untested hypothesis is that GluN2B could compensate for loss-of-function mutants of GluN2A (ref 40)”: ref 40 says that there is no sign of compensation by GluN2B.

Thank you for drawing attention to this and we agree this sentence was incorrect and not well formulated. We have removed the sentence in the revised manuscript.

- Page 4, ‘Figure 1A’ should be ‘Figure 2A’.

The figures have been reorganised and the in-text references to figures corrected in the revised manuscript.

- Page 8, ‘Table II’ should be ‘Table III’.

The tables have been reorganised and the in-text references to tables corrected in the revised manuscript.

- Page 9, “Charge transfer of mutant NMDAR-EPSCs based on the Swanger and others”: the model of Swanger predicts theoretical EPSCs mediated by GluN2A subunits only. One can reasonably expect a different outcome in real neurons expressing a mixture of receptor populations.

We have now removed the graph 5D of the original manuscript and the reference to it from the manuscript.

- Page 13, ‘that there is not complete compensation by GluN2B’, ‘no’ instead of ‘not’.

We have revised the sentence in question on page 14 line 7.

- Page 14, ‘Figure 2’ end of 2nd paragraph should be ‘Figure 1’.

We have corrected the figure reference referred to

- Methods: Is there a reason for why series resistances were not compensated for?

A conscious decision was made to not perform on-line series resistance compensation, mainly because we feared we would experience an unacceptable increase in the rate of attrition. If high levels of series resistance compensation are used (e.g. 90-95%), modest changes in access resistance during the recording protocol can cause ‘ringing’ and loss of the cell recording. This is made harder by the requirement to simultaneously monitor the access resistance in the two recording channels in which we are changing holding potential. Furthermore, changes in filtering of the EPSCs will often represent only a small fraction of the filtering resulting from the poor space clamp in these relatively large neurons. Nonetheless, we do appreciate that differences in access resistance could be adding nuisance variability and compensating for this could increase the power of our analysis.

We have reanalysed **all** of our data using off-line series resistance compensation (similar to Traynelis, 1998, and based on SeriesResistanceComp in Proc02_Apr4Web.ipf,

<http://www3.mpibpc.mpg.de/groups/neher/index.php?page=software>), which are now presented in **Figures 2, 4, 5, S1 (except (S1.3)), S4, and Table S4** of the new manuscript. The series resistance (R_s) and membrane time constant (C_m) were estimated from the current response to a square voltage pulse delivered at the start of each recording protocol, and the reversal potential was estimated from the current-voltage relationship of the (uncompensated) currents (which were just below 0 mV). The R_s and C_m values obtained for each cell at -100 mV and +20 mV were used to apply off-line series resistance compensation such that the final uncompensated series resistance of 3 Mohm. The % compensation applied was calculated as follows:

$$R_{s_{comp}}(\%) = 100 * \max \left(1 - \frac{R_{s_{final}}}{R_s}, 0 \right)$$

(The NMDA component of currents at -100 mV is likely negligible and so the current-voltage relationship of the remaining AMPA component is assumed linear between the -100 mV holding potential and the reversal potential, which simplifies the offline correction. Likewise current-voltage relationship of the NMDA current is linear between +20 mV and the reversal potential.)

Response to reviewers: COMMSBIO-20-2559-T

Overall, the series resistance compensation did not change our inferences but it did reduce variability in our estimates. However, there is a limitation of the off-line series resistance compensation we've used: it depends on a linear current-voltage relationship. The currents at many of the other holding potentials (e.g. -60, -40, -20 mV) have some NMDA component whose current-voltage relationship is non-linear between holding potential and the reversal potential – the nonlinear current-voltage relationship complicates the offline corrections. For these reasons, the fact that the results for the current voltage relationships are largely negative results, coupled with our objective to simplify the manuscript, we have removed what was Figure 4 in original manuscript. Details of the methods for the offline series resistance compensation are provided at **page 22 lines 6-21** of the updated manuscript.

- Methods: Page 22 please clarify the mathematical expressions ' $\log(\text{ratio}) \sim 0 + \text{genotype} + (1|\text{animal/slice})$ '. Also expression on top of page 23.

This expression uses notation started by Wilkinson-Rogers and developed by Pinheiro and Bates for mixed models, with the description of the formula syntax, which is described well in Bates et al. (2015) J. Stat. Softw. The syntax is (almost) standard across a number packages used for linear mixed modelling (e.g. *lme4* and *brms* in R; *statsmodels*, *Rpy2* in Python, *fitlme* in Matlab).

We will breakdown the above expression:

log(response) is the \log_e -transformed response

\sim is a symbol used to define the relationship of the outcome (e.g. $\log(\text{response})$) with the predictors (e.g. genotype and the random intercepts)

0 + indicates that we want to fit the model without an intercept. This was acceptable because the predictors were categorical and we wanted to set a prior on the coefficients, where they reflect estimates of the predictors (not the differences between the predictors and the intercept). Since we have taken a frequentist approach to model fitting in the revised manuscript (and so setting a prior is not a concern), we fit the model with an intercept so this part of the notation is omitted.

genotype represents a fixed effect (e.g. in Figure 1 this corresponds to WT, HET, HOM; in Figure 3 this corresponds to WT, C436R, T531M, R518H, K669N and L812M). Fixed factors are those whose (unknown) population mean that we are trying to estimate is a fixed quantity. In the revised manuscript we refer to this as 'mutation' in analyses relating to figures 2, 3 and 4, and 'genotype' in analyses relating to figure 5, since they more accurately describe these variables.

(1|animal/slice) represents the nested random intercept in the model, which accounts for dependence structure in the data - slices are nested within animals. The **1|** symbols indicate that the factors that follow are random factors, that is to say that their population mean is a random variable. Overall this notation is short hand for **(1|animal) + (1|animal:slice)**.

The R formulae are now provided along with the summary statistics for the fixed and random effects in supplemental data **Figures S1.1-1.11**. Further details on the statistical analysis have been included in **page 24 line 22 to page 26 line 13** in the updated manuscript, including a reference for the model formula in **page 25 lines 6-7**.

- Supplementary table II: It is unclear to what "n,slices,animals" refers to, especially when compared to the numbers indicated on the violin plots (in all figures).

We agree that this was confusing. It arises because the numbers in the table reflect the data prior to multivariate outlier detection. We have removed these tables because we plot the raw data points (post-outlier detection) in the figures of the revised manuscript. The sample sizes are clearly tabulated in Table S3 of the revised manuscript.

Again, we would like to thank both the reviewers for spending time with this manuscript and providing us with constructive and useful comments.

REVIEWERS' COMMENTS:

Reviewer #1 (Remarks to the Author):

This is a re-review of a previously reviewed manuscript. The authors followed all comments and concerns of this reviewer in their revised version.

Reviewer #2 (Remarks to the Author):

The authors have made significant efforts in their revision – in particular by providing new computational experiments, re-analysis and extensive re-writing - leading to an improved manuscript.

I have just one final concern. As rightly suggested by the other reviewer, the authors now mention in their revised manuscript unconventional GluN3-containing NMDARs. However, glycine-gated GluN1/GluN3 receptors are not exclusively juvenile receptors as stated (see revised Introduction). Indeed, diheteromeric GluN1/GluN3A receptors also form functional receptors in the adult medial habenula as recently shown by Otsu et al. (Science 2019). For correctness, the text should be amended accordingly.

Overall, the authors should be praised for a nice and thorough study.